# *Info*BRIDGE: MUTUAL INFORMATION ESTIMATION VIA BRIDGE MATCHING

**Sergei Kholkin**
Applied AI Institute, Russia
kholkinsd@gmail.com

**Ivan Butakov**
Applied AI Institute, Russia,
MIRAI*, Russia
Institute of Numerical Mathematics (RAS),
Russia
ivan.butakov@applied-ai.ru

**Evgeny Burnaev**
Applied AI Institute, Russia,
AXXX, Russia
e.burnaev@applied-ai.ru

**Nikita Gushchin**
Applied AI Institute, Russia,
AXXX, Russia
i.nikita.gushchin@gmail.com

**Alexander Korotin**
Applied AI Institute, Russia,
AXXX, Russia
iamalexkorotin@gmail.com

## ABSTRACT

Diffusion bridge models have recently become a powerful tool in the field of generative modeling. In this work, we leverage them to address another important problem in machine learning and information theory, the estimation of the mutual information (MI) between two random variables. Neatly framing MI estimation as a domain transfer problem, we construct an unbiased estimator for data posing difficulties for conventional MI estimators. We showcase the performance of our estimator on three standard MI estimation benchmarks, i.e., low-dimensional, image-based and high MI, and on real-world data, i.e., protein language model embeddings. The code for our estimator can be found at:

https://github.com/SKholkin/infobridge

## 1 INTRODUCTION

Information theory offers an extensive set of tools for quantifying probabilistic relations between random variables. It is widely used in machine learning for advanced statistical analysis (Berrett & Samworth, 2017; Sen et al., 2017; Duong & Nguyen, 2023b; Bounoua et al., 2024a), assessment of deep neural networks' performance and generalization capabilities (Tishby & Zaslavsky, 2015; Xu & Raginsky, 2017; Goldfeld et al., 2019; Steinke & Zakynthinou, 2020; Amjad et al., 2022; Butakov et al., 2024b), self-supervised and semi-supervised learning (Linsker, 1988; Bell & Sejnowski, 1995; Hjelm et al., 2019; Stratos, 2019; Bachman et al., 2019; Veličković et al., 2019; van den Oord et al., 2019; Tschannen et al., 2020) and regularization or alignment in generative modeling (Chen et al., 2016; Belghazi et al., 2018; Ardizzone et al., 2020; Wang et al., 2024).

The majority of the aforementioned applications revolve around one of the central information-theoretic quantities – *mutual information* (MI). Due to several outstanding properties, MI is widely used as an invariant measure of non-linear dependence between random variables. Unfortunately, recent studies suggest that the curse of dimensionality is highly pronounced when estimating MI (Goldfeld et al., 2020; McAllester & Stratos, 2020). Additionally, it is argued that long tails, high values of MI and some other particular features of complex probability distributions can make mutual information estimation even more challenging (Czyż et al., 2023). On the other hand, recent developments in neural estimation methods demonstrate that sophisticated parametric estimators can achieve notable practical success in situations where traditional mutual information estimation techniques struggle (Belghazi et al., 2018; van den Oord et al., 2019; Song & Ermon, 2020; Rhodes et al., 2020; Ao & Li, 2022; Butakov et al., 2024a). Among neural estimators, generative approaches

---

*Moscow Independent Research Institute of Artificial Intelligence

are of particular interest, as they have proven to be effective in handling complex data (Duong & Nguyen, 2023a; Franzese et al., 2024; Butakov et al., 2024a). Since MI estimation is closely tied to approximation of a joint probability distribution, one can argue that leveraging state-of-the-art generative models, e.g., diffusion models, may result in additional performance gains.

**Diffusion Bridge Matching.** Diffusion models are a powerful type of generative models that show an impressive quality of image generation (Ho et al., 2020; Rombach et al., 2022). However, they have some disadvantages, such as the inability to perform data-to-data translation via diffusion. To tackle this problem, a novel promising approach based on Reciprocal Processes (Léonard et al., 2014) and Schrödinger Bridges (Schrödinger, 1932; Léonard, 2014a) has emerged. This approach is called the *diffusion bridge matching* and is used for learning generative diffusion processes for data-to-data translation. This type of model has shown itself as a powerful approach for numerous applications in biology (Tong et al., 2024; Bunne et al., 2023), chemistry (Somnath et al., 2023; Igashov et al., 2024), computer vision (Liu et al., 2023; Shi et al., 2023; Zhou et al., 2024), speech processing (Chen et al., 2023), unpaired learning (Shi et al., 2023; Gushchin et al., 2024b; 2023b;a; Ksenofontov & Korotin, 2025; Kholkin et al., 2026; Carrasco et al., 2026) and beyond (He et al., 2024; Gushchin et al., 2025).

**Contributions.** In this work, we employ the Diffusion Bridge Matching for the MI estimation.

1. **Theory.** We propose an unbiased mutual information estimator based on reciprocal processes, their diffusion representations and the Girsanov theorem (§4.1).
2. **Practice.** Building on the proposed theoretical framework and the powerful generative methodology of diffusion bridges, we develop a practical algorithm for MI estimation, named *Info*Bridge (§4.2). We demonstrate that our method achieves performance comparable to existing approaches on low-dimensional benchmarks and superior performance on image data benchmarks, protein embeddings data benchmark and high MI benchmark (§5).

**Notations.** We work in $\mathbb{R}^D$, which is the $D$-dimensional Euclidean space equipped with the Euclidean norm $\|\cdot\|$. We use $\mathcal{P}(\mathbb{R}^D)$ to denote the absolutely continuous Borel probability distributions whose variance and differential entropy are finite. To denote the density of $q \in \mathcal{P}(\mathbb{R}^D)$ at a point $x \in \mathbb{R}^D$, we use $q(x)$. We write KL $(\cdot\|\cdot)$ to denote the Kullback-Leibler divergence between two distributions. We denote the Dirac delta function as $\delta$. We use $\Omega$ to denote the space of trajectories, i.e., continuous $\mathbb{R}^D$-valued functions of $t \in [0,1]$. We write $\mathcal{P}(\Omega)$ to denote the probability distributions on the trajectories $\Omega$ whose marginals at $t = 0$ and $t = 1$ belong to $\mathcal{P}(\mathbb{R}^D)$; this is the set of stochastic processes. We use $dW_t$ to denote the differential of the standard Wiener process $W \in \mathcal{P}(\Omega)$. We use $Q_{|x_0}$ and $Q_{|x_0,x_1}$ to denote the distribution of stochastic process $Q$ conditioned on $Q$'s values $x_0$ and $x_0, x_1$ at times $t = 0$ and $t = 0, 1$, respectively. For a process $Q \in \mathcal{P}(\Omega)$, we denote its marginal distribution at time $t$ by $q(x_t) \in \mathcal{P}(\mathbb{R}^D)$, and if the process is conditioned on its value $x_s$ at time $s$, we denote the marginal distribution of such a process at time $t$ by $q(x_t|x_s) \in \mathcal{P}(\mathbb{R}^D)$. SDE states for Stochastic Differential Equation (Øksendal, 2003, §5).

## 2 BACKGROUND

**Mutual information.** Information theory is a well-established framework for analyzing and quantifying interactions between random vectors. In this framework, mutual information (MI) serves as a fundamental and invariant measure of the non-linear dependence between two $\mathbb{R}^D$-valued random vectors $X_0, X_1$. It is defined as follows:

$$I(X_0; X_1) \stackrel{\text{def}}{=} \text{KL}\left(\Pi_{X_0,X_1} \| \Pi_{X_0} \otimes \Pi_{X_1}\right), \tag{1}$$

where $\Pi_{X_0,X_1}$ and $\Pi_{X_0}, \Pi_{X_1}$ are the joint and marginal distributions of a pair of random vectors $(X_0, X_1)$. If the corresponding PDF $\pi(x_0, x_1)$ exists, the following also holds:

$$I(X_0; X_1) = \mathbb{E}_{\pi(x_0,x_1)} \log \frac{\pi(x_0, x_1)}{\pi(x_0)\pi(x_1)}. \tag{2}$$

Mutual information is symmetric, non-negative and equals zero if and only if $X_0$ and $X_1$ are independent. MI is also invariant to bijective mappings: $I(X_0; X_1) = I(g(X_0); X_1)$ if $g^{-1}$ exists and $g, g^{-1}$ are measurable (Cover & Thomas, 2006; Polyanskiy & Wu, 2024).

**Brownian Bridge**. Let $W^\epsilon$ be the Wiener process with a constant volatility $\epsilon > 0$, i.e., it is described by the SDE $dW^\epsilon = \sqrt{\epsilon}dW_t$, where $W_t$ is the standard Wiener process. Let $W^\epsilon_{|x_0,x_1}$ denote the

process $W^\epsilon$ conditioned on its values $x_0, x_1$ at times $t = 0, 1$, respectively. This process $W^\epsilon_{|x_0,x_1}$ is called the Brownian Bridge (Ibe, 2013, Chapter 9). o

**Reciprocal processes.**   Reciprocal processes are a class of stochastic processes that have recently gained attention of research community in the contexts of stochastic optimal control (Léonard et al., 2014), Schrödinger Bridges (Schrödinger, 1932; Léonard, 2014a) and diffusion generative modeling (Liu et al., 2023; Gushchin et al., 2024a). In our paper, we consider a *particular case* of reciprocal processes which are induced by the Brownian Bridge $W^\epsilon_{|x_0,x_1}$.

Consider a joint distribution $\pi(x_0, x_1) \in \mathcal{P}(\mathbb{R}^{D\times 2})$ and define the process $Q_\pi \in \mathcal{P}(\Omega)$ as a mixture of Brownian bridges $W^\epsilon_{|x_0,x_1}$ with weights $\pi(x_0, x_1)$:

$$Q_\pi \stackrel{\text{def}}{=} \int W^\epsilon_{|x_0,x_1} d\pi(x_0, x_1).$$

This implies that to get trajectories of $Q_\pi$ one has to first sample the start and end points, $x_0$ and $x_1$, at times $t = 0$ and $t = 1$ from $\pi(x_0, x_1)$ and then simulate the Brownian Bridge $W^\epsilon_{|x_0,x_1}$. Notice that process $Q_\pi$ depends on $\epsilon$, but next in the paper we do not write $\epsilon$ and just stick to $Q_\pi$ notation. Due to the non-causal nature of trajectory formation, such a process is, in general, non markovian. The set of all mixtures of Brownian Bridges can be described as:

$$\left\{ Q \in \mathcal{P}(\Omega) \ s.t. \ \exists \pi \in \mathcal{P}(\mathbb{R}^{D\times 2}) : Q = Q_\pi \right\}$$

and is called the set of *reciprocal processes* (for $W^\epsilon$).

**Reciprocal processes conditioned on the point.**   Consider a reciprocal process $Q_\pi$ conditioned on some start point $x_0$. Let the resulting process be denoted as $Q_{\pi|x_0}$, which remains reciprocal. Then process $Q_{\pi|x_0}$ is known as the Schrödinger Föllmer process (Vargas et al., 2023). While $Q_\pi$, in general, not markovian, $Q_{\pi|x_0}$ **is markovian**. Furthermore, if $\int_{\mathbb{R}^D} \|x_1\| d\pi(x_1) < \infty$, it is a diffusion process governed by the following SDE, i.e., see Proposition A.3:

$$Q_{\pi|x_0} : dx_t = v_{x_0}(x_t, t)dt + \sqrt{\epsilon}dW_t, x_0 \sim \delta(x_0),$$

$$v_{x_0}(x_t, t) = \mathbb{E}_{q_\pi(x_1|x_t,x_0)} \left[ \frac{x_1 - x_t}{1 - t} \right]. \tag{3}$$

**Representations of reciprocal processes.** The process $Q_\pi$ can be naturally represented as a mixture of processes $Q_{\pi|x_0}$ conditioned on their starting points $x_0$:

$$Q_\pi = \int Q_{\pi|x_0} d\pi(x_0).$$

Therefore, one may also express $Q_\pi$ via an SDE but with non-markovian drift (conditioned on $x_0$):

$$Q_\pi : dx_t = v(x_t, t, x_0)dt + \sqrt{\epsilon}dW_t, x_0 \sim \pi(x_0),$$

$$v(x_t, t, x_0) = v_{x_0}(x_t, t) = \mathbb{E}_{q_\pi(x_1|x_t,x_0)} \left[ \frac{x_1 - x_t}{1 - t} \right]. \tag{4}$$

**Conditional Bridge Matching.**   Although the drift $v_{x_0}(x_t, t)$ of $Q_{\pi|x_0}$ in (3) admits a closed form, it usually cannot be computed or estimated directly due to the unavailability of a way to easily sample from $\pi(x_1|x_t, x_0)$. However, it can be recovered by solving the following regression problem (Zhou et al., 2024):

$$v_{x_0} = \arg\min_u \mathbb{E}_{q_\pi(x_1,x_t|x_0),U_{[0,1)}(t)} \left\| \frac{x_1 - x_t}{1 - t} - u(x_t, t) \right\|^2, \tag{5}$$

where $U_{[0,1]}(t)$ denotes the uniform distribution over $t \in [0, 1]$, and the optimization is performed over drift functions $u : \mathbb{R}^D \times [0, 1] \to \mathbb{R}^D$. The same holds for the $Q_\pi$ and its drift $v(x_t, t, x_0)$ through the addition of expectation w.r.t. $\pi(x_0)$:

$$v = \arg\min_u \mathbb{E}_{q_\pi(x_1,x_t|x_0)\pi(x_0),U_{[0,1)}(t)} \left\| \frac{x_1 - x_t}{1 - t} - u(x_t, t, x_0) \right\|^2 = \tag{6}$$

$$\arg\min_{u} \mathbb{E}_{q_\pi(x_1, x_t, x_0), U_{[0,1)}(t)} \left\| \frac{x_1 - x_t}{1 - t} - u(x_t, t, x_0) \right\|^2, \tag{7}$$

where $u : \mathbb{R}^D \times [0,1) \times \mathbb{R}^D \to \mathbb{R}^D$. Problem (7) is usually solved with standard deep learning techniques. Namely, one parametrizes $u$ with a neural network $v_\theta$, and minimizes (7) using stochastic gradient descent and samples drawn from $q_\pi(x_0, x_t, x_1)$. The latter sampling is easy if one can sample from $\pi(x_0, x_1)$. Indeed, $q_\pi(x_0, x_t, x_1) = q_\pi(x_t|x_0, x_1)\pi(x_0, x_1)$, and one can sample first from $\pi(x_0, x_1)$ and then from $q_\pi(x_t|x_0, x_1)$, which is the time slice of the Brownian Bridge (Korotin et al., 2024, Eq 14), i.e., $W^\epsilon_{|x_0, x_1}(\cdot|t)$.

Such a procedure of learning drift $v$ with a neural network is popular in generative modeling to solve a problem of sampling from conditional distribution $\pi(x_1|x_0)$ and is frequently applied in the image-to-image transfer (Liu et al., 2023). The procedure of learning drift $v(x_t, t, x_0)$ (7) is usually called the *conditional* (or augmented) *bridge matching* (De Bortoli et al., 2023; Zhou et al., 2024). In addition, such procedure can also be derived through the well-celebrated Doob $h$-transform (De Bortoli et al., 2023; Palmowski & Rolski, 2002) or reversing a diffusion (Zhou et al., 2024).

## 3 RELATED WORK

**Mutual information estimators.** Mutual information estimators fall into two main categories: *non-parametric* and *parametric*. Parametric estimators are also subdivided into *discriminative* and *generative* (Song & Ermon, 2020; Federici et al., 2023). Beyond this natural categorization, we also distinguish *diffusion-based* approaches to better position our method w.r.t. prior work.

**Non-parametric estimators.** Classical MI estimation methods use non-parametric density estimators, such as kernel density estimation (Weglarczyk, 2018; Goldfeld et al., 2019) and $k$-nearest neighbors (Kozachenko & Leonenko, 1987; Kraskov et al., 2004; Berrett et al., 2019). The estimated densities are plugged into (2), and MI is computed via Monte Carlo integration. While appealing in low-dimensional settings, these approaches have been shown to fail on high-dimensional or complex data (Goldfeld et al., 2019, §5.3; Czyż et al., 2023, §6.2; Butakov et al., 2024a, Table 1).

**Non-diffusion-based generative estimators.** Parametric density models—such as normalizing flows and variational autoencoders—can be used to estimate MI via density estimation (Song & Ermon, 2020; McAllester & Stratos, 2020; Ao & Li, 2022; Duong & Nguyen, 2023a). However, results in (Song & Ermon, 2020, Figures 1,2) show that direct PDF estimation may introduce significant bias. To address this, recent works (Duong & Nguyen, 2023a; Butakov et al., 2024a; Dahlke & Pacheco, 2025) avoid PDF estimation and instead estimate the density ratio in (2), leveraging the invariance property of MI. While these methods perform better on synthetic benchmarks, they may still suffer from inductive biases introduced by simplified closed-form approximations.

**Discriminative estimators.** Another class of MI estimators relies on training a classifier to distinguish between samples from the joint distribution $\pi(x_0, x_1)$ and the product of marginals $\pi(x_0)\pi(x_1)$, as in MINE (Belghazi et al., 2018), InfoNCE (van den Oord et al., 2019), fDIME (Letizia et al., 2024) and SMILE (Song & Ermon, 2020). These approaches use variational bounds on KL divergence and offer parametric estimators suitable for high-dimensional and complex data. Despite their scalability, they suffer from well-known theoretical limitations, such as high variance in MINE and large batch size requirements in InfoNCE (Song & Ermon, 2020). Recent high-MI and complex distributions based benchmarks further indicate that discriminative methods may underperform compared to generative approaches (Franzese et al., 2024; Butakov et al., 2024a).

**Neural Diffusion Estimator for MI (MINDE).** One of the most recent generative methods for MI Estimation is diffusion-based (Song et al., 2021) MINDE (Franzese et al., 2024). To estimate $\text{KL}\left(\pi^A \| \pi^B\right)$ the authors learn two standard backward diffusion models to generate data from distributions $\pi^A$ and $\pi^B$, e.g., for $\pi^A$:

$$Q^A : \underbrace{dx_t = [-f(x_t, t) + g(t)^2 s^A(x_t, t)]dt + g(t)d\hat{W}_t}_{\text{backward diffusion}}, \quad x_T \sim q_T^A(x_T), \tag{8}$$

where $f$ and $g$ are the drift and volatility coefficients, respectively, of the forward diffusion (Song et al., 2021), $d\hat{W}_t$ is the Wiener process when time flows backwards, and $q_t^A$ is the distribution of the noised data at time $t$ (Franzese et al., 2024, §2, 3). The similar expressions hold for $\pi^B$ and

$Q^B$. Then, the authors formulate a KL divergence estimator through the difference of diffusion *score functions*:

$$\text{KL}\left(\pi^A\big\|\pi^B\right) = \text{KL}\left(Q^A\big\|Q^B\right) = \int_0^T \mathbb{E}_{q_t^A(x_t)}\left[\frac{g(t)^2}{2}\|s^A(x_t,t) - s^B(x_t,t)\|^2\right]dt + \text{KL}\left(q_T^A\big\|q_T^B\right) \tag{9}$$

Here, $\text{KL}\left(q_T^A\big\|q_T^B\right)$ is the **bias** term, which vanishes only when diffusion has infinitely many steps, i.e., $T \to \infty$. When the diffusion score functions $s^A$ and $s^B$ (8) are properly learned, one can draw samples from the forward diffusion $q_t^A(x_t) = q_t^A(x_t|x_0)q(x_0)$ and compute the estimate of KL divergence (9). In this way, the authors transform the problem of training the KL divergence estimator into the problem of learning the backward diffusions (8) that generate *data from noise*.

To estimate **mutual information**, the authors propose a total of four equivalent methods, all based on the estimation of up to three KL divergences (9) or their expectations.

# 4 *Info*BRIDGE MUTUAL INFORMATION ESTIMATOR

In (§4.1), we propose our novel MI estimator which is based on difference of diffusion drifts of conditional reciprocal processes. We explain the practical learning procedure in (§4.2) and suggest some straightforward generalizations of our method in (§4.3).

## 4.1 COMPUTING MI THROUGH RECIPROCAL PROCESSES

Consider the problem of MI estimation for random variables $X_0$ and $X_1$ with joint distribution $\pi(x_0, x_1)$. To tackle this problem, we employ reciprocal processes:

$$Q_\pi \stackrel{\text{def}}{=} \int W_{|x_0,x_1}^\epsilon d\pi(x_0,x_1), Q_\pi^{\text{ind}} \stackrel{\text{def}}{=} \int W_{|x_0,x_1}^\epsilon d\pi(x_0)d\pi(x_1). \tag{10}$$

We show that the KL between the distributions $\pi(x_0, x_1)$ and $\pi(x_0)\pi(x_1)$ (1) is equal to the KL between the reciprocal processes $Q_\pi$ and $Q_\pi^{\text{ind}}$, and decompose the latter into the difference of drifts.

First, let us introduce a classical stochastic calculus result that allows us to compute the KL between diffusion processes, e.g., $Q_\pi$ and $Q_\pi^{\text{ind}}$ SDE representations.

**KL divergence between diffusion processes.** Consider two diffusion processes with the same volatility coefficient $\sqrt{\epsilon}$ that start at the same distribution $\pi_0$:

$$Q^\alpha: \ dx_t = f^\alpha(x_t,t)\,dt + \sqrt{\epsilon}\,dW_t, \quad x_0 \sim \pi_0(x_0), \qquad \alpha \in \{A, B\}. \tag{11}$$

By the application of the disintegration theorem (Léonard, 2014b, §1) and the Girsanov theorem (Léonard, 2012) one can derive the KL divergence between these diffusions:

$$\text{KL}\left(Q^A\big\|Q^B\right) = \frac{1}{2\epsilon}\int_0^1 \mathbb{E}_{q^A(x_t)}\left[\|f^A(x_t,t) - f^B(x_t,t)\|^2\right]dt, \tag{12}$$

where $q^A(x_t)$ is the marginal distribution of $Q^A$ at time $t$.

This allows one to estimate the KL divergence between two diffusions with the same volatility coefficient and the same initial distributions, knowing only their *drifts* and marginal samples $x_t \sim q^A(x_t)$. This fact is widely used in Bridge Matching (Shi et al., 2023; Peluchetti, 2023), Diffusion (Franzese et al., 2024) and Schrödinger Bridge Models (Vargas et al., 2021; Gushchin et al., 2023a).

**Theorem 4.1** (Mutual Information decomposition). *Consider random variables $X_0, X_1$ and their joint distribution $\pi(x_0, x_1)$, such that $I(X_0; X_1) < \infty$ and $\int_{\mathbb{R}^D} \|x_1\|\,d\pi(x_1) < \infty$. Consider reciprocal processes $Q_\pi$, $Q_\pi^{\text{ind}}$ induced by distributions $\pi(x_0, x_1)$ and $\pi(x_0)\pi(x_1)$, respectively, as in (10). Then the MI between the random variables $X_0$ and $X_1$ can be expressed as:*

$$I(X_0; X_1) = \frac{1}{2\epsilon}\int_0^1 \mathbb{E}_{q_\pi(x_t,x_0)}\|v_{joint}(x_t,t,x_0) - v_{ind}(x_t,t,x_0)\|^2 dt, \tag{13}$$

*where*

$$v_{joint}(x_t,t,x_0) = \mathbb{E}_{q_\pi(x_1|x_t,x_0)}\left[\frac{x_1 - x_t}{1 - t}\right], \tag{14}$$

$$v_{ind}(x_t,t,x_0) = \mathbb{E}_{q_\pi^{ind}(x_1|x_t,x_0)}\left[\frac{x_1 - x_t}{1 - t}\right]. \tag{15}$$

*$v_{joint}$ and $v_{ind}$ are the drifts of the SDE representations (4) of the reciprocal processes $Q_\pi$ and $Q_\pi^{ind}$.*

---

**Algorithm 1:** *Info*Bridge. Training the model.

---

**Input** : Distribution $\pi(x_0, x_1)$ accessible by samples, initial neural network parametrization $v_\theta$ of drift functions

**Output** : Learned neural network $v_\theta$ approximating drifts $v_{\text{joint}}$ and $v_{\text{ind}}$

**repeat**

    Sample batch of pairs $\{x_0^n, x_1^n\}_{n=0}^N \sim \pi(x_0, x_1)$;

    Sample random permutation $\{\hat{x}_1^n\}_{n=0}^N = \text{Permute}(\{x_1^n\}_{n=0}^N)$;

    Sample batch $\{t^n\}_{n=1}^N \sim U[0,1]$;

    Sample batch $\{x_t^n\}_{n=1}^N \sim W_{|x_0,x_1}^\epsilon$ ;        // `Sample trajectory points from Joint.`

    Sample batch $\{\hat{x}_t^n\}_{n=1}^N \sim W_{|x_0,\hat{x}_1}^\epsilon$ ;        // `Sample trajectory point from Independent.`

    $\mathcal{L}_\theta^1 = \frac{1}{N} \sum_{n=1}^N \|v_\theta(x_t^n, t^n, x_0^n, 1) - \frac{x_1^n - x_t^n}{1 - t^n}\|^2$;

    $\mathcal{L}_\theta^2 = \frac{1}{N} \sum_{n=1}^N \|v_\theta(\hat{x}_t^n, t^n, x_0^n, 0) - \frac{\hat{x}_1^n - \hat{x}_t^n}{1 - t^n}\|^2$;

    Update $\theta$ using $\frac{\partial \mathcal{L}_\theta^1}{\partial \theta} + \frac{\partial \mathcal{L}_\theta^2}{\partial \theta}$;

**until** *converged*;

---

**Algorithm 2:** *Info*Bridge. MI estimator.

---

**Input** : Distribution $\pi(x_0, x_1)$ accessible by samples, neural network parametrization $v_\theta$ of drift functions approximating optimal drifts $v_{\text{joint}}$ and $v_{\text{ind}}$, number of samples $N$

**Output** : Mutual information estimation $\widehat{\text{MI}}$

Sample batch of pairs $\{x_0^n, x_1^n\}_{n=1}^N \sim \pi(x_0, x_1)$;

Sample batch $\{t^n\}_{n=1}^N \sim U_{[0,1)}(t)$;

Sample batch $\{x_t^n\}_{n=1}^N \sim W_{|x_0,x_1}^\epsilon(\cdot|\{t^n\}_{n=1}^N)$;

$\widehat{\text{MI}} \leftarrow \frac{1}{2\epsilon N} \sum_{n=0}^N \|v_\theta(x_t^n, t^n, x_0^n, 1) - v_\theta(x_t^n, t^n, x_0^n, 0)\|^2$

---

*Proof sketch*: Starting from the quantity $\text{KL}\left(Q_\pi \| Q_\pi^{\text{ind}}\right)$, we first apply the disintegration theorem to express it as $\mathbb{E}_{\pi(x_0)}[\text{KL}\left(Q_{\pi|x_0} \| Q_{\pi|x_0}^{\text{ind}}\right)]$ and apply it one more time to express it as $\text{KL}\left(\pi(x_0, x_1) \| \pi(x_0)\pi(x_1)\right)$. Finally, we decompose the KL between the conditional diffusion processes $\text{KL}\left(Q_{\pi|x_0} \| Q_{\pi|x_0}^{\text{ind}}\right)$ using Girsanov's theorem. Full proof is provided in Appendix A.

Once the drifts $v_{\text{joint}}$ and $v_{\text{ind}}$ are known, our Theorem 4.1 provides a straightforward way to estimate the mutual information between the random variables $X_0$ and $X_1$ by evaluating the difference between the drifts $v_{\text{joint}}(x_t, t, x_0)$ (14) and $v_{\text{ind}}(x_t, t, x_0)$ (15) at points $x_0, x_t$ sampled from the distribution of the reciprocal process $Q_\pi$ at times $0, t$. Regularity assumptions (Shi et al., 2023, Appendix C) are relatively mild and common for diffusion bridges generative modeling, i.e., they include restrictions such as finite first moment.

**Remark**. Our approach is different from MINDE, because we frame MI estimation as a **domain transfer** task, while MINDE frames it as **generative modeling** task. Our formulation allows **unbiased**[1] MI estimation (13) while MINDE has non zero bias term (9). Furthermore, the domain transfer perspective yields easier to learn trajectories and reduced variance in MI estimation Tables 2 and 3. More details on the comparison with MINDE are presented in the Appendix C.

### 4.2 *Info*BRIDGE. PRACTICAL OPTIMIZATION PROCEDURE

The drifts $v_{\text{joint}}$ and $v_{\text{ind}}$ of reciprocal processes $Q_\pi$ and $Q_\pi^{\text{ind}}$ can be recovered by the conditional Bridge Matching procedure, see (7). We have to solve optimization problem (7) by parametrizing $v_{\text{joint}}$ and $v_{\text{ind}}$ with neural networks $v_{\text{joint},\phi}$ and $v_{\text{ind},\psi}$, respectively, and applying Stochastic Gradient Descent on Monte Carlo approximation of (7). The sampling from the distribution $q_\pi(x_t, x_0)$ of reciprocal process $Q_\pi$ at times $0, t$ is easy because:

$$q_\pi(x_t, x_0) = \mathbb{E}_{q_\pi(x_1)}[q_\pi(x_t, x_0|x_1)] = \mathbb{E}_{q_\pi(x_1)}[q_\pi(x_t|x_1, x_0)\pi(x_0|x_1)].$$

---

[1] In our case, by the word *unbiased* we mean that under full access to distributions and under the assumption that we can learn drift functions ideally, the estimation of the objective is unbiased.

Table 1: Mean MI estimates over 10 seeds using 10k test samples against ground truth (GT), adopted from Franzese et al. (2024). The closer the estimate is to the ground truth, the better. Color indicates relative negative (red) and positive bias (blue).

| | Method Type | | | | | | | | | | | | | | | | | | | | | | | | | | | | | | | | | | | | | | | | | | |
|---|---|---|---|---|---|---|---|---|---|---|---|---|---|---|---|---|---|---|---|---|---|---|---|---|---|---|---|---|---|---|---|---|---|---|---|---|---|---|---|---|---|---|---|
| GT | | 0.2 | 0.4 | 0.3 | 0.4 | 0.4 | 0.4 | 0.4 | 1.0 | 1.0 | 1.0 | 1.0 | 0.3 | 1.0 | 1.3 | 1.0 | 0.4 | 1.0 | 0.6 | 1.6 | 0.4 | 1.0 | 1.0 | 1.0 | 1.0 | 1.0 | 1.0 | 1.0 | 1.0 | 0.2 | 0.4 | 0.2 | 0.3 | 0.2 | | 0.4 | 0.3 | 0.4 | 1.7 | 0.3 | 0.4 |
| *Info*Bridge(ours) | Bridge Matching | 0.3 | 0.5 | 0.3 | 0.4 | 0.4 | 0.4 | 0.4 | 0.9 | 1.0 | 1.0 | 1.0 | 0.3 | 1.0 | 1.3 | 1.0 | 0.4 | 1.0 | 0.6 | 1.7 | 0.4 | 1.0 | 1.0 | 1.0 | 1.0 | 0.9 | 0.9 | 1.0 | 1.0 | 0.0 | 0.0 | 0.2 | 0.3 | 0.2 | | 0.5 | 0.3 | 0.5 | 1.3 | 0.4 | 0.4 |
| NVF | Flow | 0.2 | 0.4 | 0.3 | 0.6 | 0.4 | 0.4 | 0.4 | 1.0 | 1.0 | 1.0 | 1.0 | 0.3 | 1.0 | 1.3 | 1.0 | 0.4 | 1.0 | 0.6 | 1.5 | 0.4 | 1.0 | 1.0 | 1.0 | 0.8 | 0.5 | 0.6 | 0.9 | 1.0 | ∞ | ∞ | ∞ | ∞ | 0.5 | 0.2 | -0.4 | 0.4 | 0.2 | 1.5 | 0.2 | 0.4 |
| JVF | | 0.0 | 0.0 | 0.0 | 0.0 | 0.0 | 0.4 | 0.4 | 1.0 | 1.0 | 1.0 | 1.0 | 0.3 | 1.0 | 1.3 | 1.0 | 0.4 | 1.0 | 0.6 | 1.6 | 0.4 | 1.0 | 1.0 | 1.0 | 0.8 | 0.3 | 0.4 | 1.0 | 0.9 | 0.9 | 1.8 | 2.7 | 0.0 | 0.1 | 1.0 | 0.1 | 0.0 | 0.0 | 1.6 | 0.2 | 0.4 |
| MINDE–J | Diffusion | 0.2 | 0.4 | 0.3 | 0.4 | 0.4 | 0.4 | 0.4 | 1.2 | 1.0 | 1.0 | 1.0 | 0.3 | 1.0 | 1.3 | 1.0 | 0.4 | 1.0 | 0.6 | 1.7 | 0.4 | 1.1 | 1.0 | 1.0 | 1.0 | 0.9 | 0.9 | 1.1 | 1.0 | 1.0 | 0.1 | 0.2 | 0.2 | 0.3 | 0.2 | 0.5 | 0.3 | 0.4 | 1.7 | 0.3 | 0.4 |
| MINDE–C | | 0.2 | 0.4 | 0.3 | 0.4 | 0.4 | 0.4 | 0.4 | 1.0 | 1.0 | 1.0 | 1.0 | 0.3 | 1.0 | 1.3 | 1.0 | 0.4 | 1.0 | 0.6 | 1.6 | 0.4 | 1.0 | 1.0 | 1.0 | 0.9 | 0.9 | 0.9 | 1.0 | 1.0 | 1.0 | 0.3 | 0.2 | 0.3 | 0.2 | | 0.4 | 0.3 | 0.4 | 1.7 | 0.3 | 0.4 |
| MINE | Classical | 0.2 | 0.4 | 0.2 | 0.4 | 0.4 | 0.4 | 0.4 | 1.0 | 1.0 | 1.0 | 1.0 | 0.3 | 1.0 | 1.3 | 1.0 | 0.4 | 1.0 | 0.6 | 1.6 | 0.4 | 0.9 | 0.9 | 0.9 | 0.8 | 0.7 | 0.6 | 0.9 | 0.9 | 0.9 | 0.0 | 0.0 | 0.1 | 0.1 | 0.1 | 0.2 | 0.2 | 0.4 | 1.7 | 0.3 | 0.4 |
| InfoNCE | | 0.2 | 0.4 | 0.3 | 0.4 | 0.4 | 0.4 | 0.4 | 1.0 | 1.0 | 1.0 | 1.0 | 0.3 | 1.0 | 1.3 | 1.0 | 0.4 | 1.0 | 0.6 | 1.6 | 0.4 | 0.9 | 1.0 | 0.8 | 0.8 | 0.8 | 0.9 | 1.0 | 1.0 | 0.0 | 0.2 | 0.3 | 0.2 | 0.3 | 0.2 | 0.4 | 0.3 | 0.4 | 1.7 | 0.3 | 0.4 |
| DoE (Gaussian) | | 0.2 | 0.5 | 0.3 | 0.6 | 0.4 | 0.4 | 0.4 | 0.7 | 1.0 | 1.0 | 0.4 | 0.7 | 7.8 | 1.0 | 0.6 | 0.9 | 1.3 | | 0.4 | 0.7 | 1.0 | 1.0 | 0.5 | 0.6 | 0.6 | 0.6 | 0.7 | 0.8 | 6.7 | 7.9 | 1.8 | 2.5 | 0.6 | 4.2 | 1.2 | 1.6 | 0.1 | 0.4 | | |
| DoE (Logistic) | | 0.1 | 0.4 | 0.2 | 0.4 | 0.4 | 0.4 | 0.4 | 0.6 | 0.9 | 1.0 | 0.3 | 0.7 | 7.8 | 1.0 | 0.6 | 0.9 | 1.3 | | 0.4 | 0.8 | 1.1 | 1.0 | 0.5 | 0.6 | 0.6 | 0.7 | 0.8 | 0.8 | 0.5 | 0.8 | 1.5 | 0.6 | 1.6 | 0.1 | 0.4 | | | | | |
| KSG | | 0.2 | 0.4 | 0.2 | 0.2 | 0.4 | 0.4 | 0.4 | 0.2 | 0.9 | 0.7 | 1.0 | 0.3 | 0.2 | 1.1 | 1.0 | 0.4 | 0.7 | 0.6 | 1.3 | 0.4 | 0.2 | 0.9 | 0.7 | 0.2 | 0.7 | 0.6 | 0.2 | 0.9 | 0.7 | 0.2 | 0.2 | 0.1 | 0.1 | 0.1 | 0.2 | 0.2 | 0.4 | 1.7 | 0.3 | 0.4 |

Column labels (left to right): *Asinh @ St 1 × 1 (dof=1)*, *Asinh @ St 2 × 2 (dof=1)*, *Asinh @ St 3 × 3 (dof=1)*, *Asinh @ St 5 × 5 (dof=2)*, *Asinh @ St 5 × 5 (dof=1)*, *Bimodal 1 × 1*, *Bivariate Nm 1 × 1*, *Ht @ Bivariate Nm 1 × 1*, *Ht @ Mn 25 × 25 (2-pair)*, *Ht @ Mn 3 × 3 (2-pair)*, *Ht @ Mn 5 × 5 (2-pair)*, *Mn 2 × 2 (2-pair)*, *Mn 2 × 2 (dense)*, *Mn 25 × 25 (2-pair)*, *Mn 25 × 25 (dense)*, *Mn 3 × 3 (2-pair)*, *Mn 3 × 3 (dense)*, *Mn 5 × 5 (2-pair)*, *Mn 5 × 5 (dense)*, *Mn 50 × 50 (dense)*, *Nm CDF @ Bivariate Nm 1 × 1*, *Nm CDF @ Mn 25 × 25 (2-pair)*, *Nm CDF @ Mn 3 × 3 (2-pair)*, *Nm CDF @ Mn 5 × 5 (2-pair)*, *Sp @ Mn 25 × 25 (2-pair)*, *Sp @ Mn 3 × 3 (2-pair)*, *Sp @ Mn 5 × 5 (2-pair)*, *Sp @ Nm CDF @ Mn 25 × 25 (2-pair)*, *Sp @ Nm CDF @ Mn 3 × 3 (2-pair)*, *Sp @ Nm CDF @ Mn 5 × 5 (2-pair)*, *St 1 × 1 (dof=1)*, *St 2 × 2 (dof=1)*, *St 2 × 2 (dof=2)*, *St 3 × 3 (dof=2)*, *St 5 × 5 (dof=2)*, *Swiss roll 2 × 1*, *Uniform 1 × 1 (additive noise=0.1)*, *Uniform 1 × 1 (additive noise=0.75)*, *Wiggly @ Bivariate Nm 1 × 1*

Therefore, to sample from $q_\pi(x_t, x_0)$ it suffices to sample $x_0, x_1 \sim \pi(x_0, x_1)$ and sample from $q_\pi(x_t|x_1, x_0)$ which is again just a Brownian Bridge.

**Vector field parametrization.** In practice, we replace two separate neural networks that approximate the drifts $v_{\text{joint}}(x_t, t, x_0)$ and $v_{\text{ind}}(x_t, t, x_0)$ with a single neural network that incorporates an additional binary input. Specifically, we introduce a binary input $s \in \{0, 1\}$ to unify the drift approximations in the following way: $v_\theta(\cdot, 1) \approx v_{\text{joint}}(\cdot)$ and $v_\theta(\cdot, 0) \approx v_{\text{ind}}(\cdot)$. Such binary conditioning is widely used for the conditioning of diffusion (Ho & Salimans, 2021) and bridge matching (Bortoli et al., 2024) models. In Appendix C.5 we show that it provides a more accurate estimation.

We call our practical MI estimation algorithm *Info*Bridge. The drifts $v_\theta$ training procedure is described in Algorithm 1 and the MI estimation procedure is described in Algorithm 2. One can see that both model training and MI estimation procedures are simulation-free.

Several strategies exist for sampling data used in computing the independent drift loss $\mathcal{L}_\theta^2$ in Algorithm 1. While regular permutation works well in practice, it may introduce correlations (Letizia et al., 2024). Alternatives include "derangement" permutations (Letizia et al., 2024) or resampling from the marginal. See Appendix C.4 for further discussion.

The computational complexity of our algorithm as a Diffusion Bridge model is greater than that of discriminative methods and is similar to that of MINDE.

### 4.3 GENERALIZATIONS AND IMPLICATIONS

**Generalizations.** Our method admits several straightforward extensions. For completeness, we present a method for the unbiased estimation of the general KL divergence in Appendix B.1. According to Theorem B.1 the KL divergence between any two distributions can be decomposed into the difference of diffusion drifts in a similar way to (13).

Furthermore, our method can be extended to estimation of MI between random variables of different dimensionality (Appendix B.5) and naturally generalizes to the estimation of mutual information involving more than two random variables, known as interaction information (Appendix B.4). Practical procedures for these generalizations can also be derived in a similar way to (§4.2).

**Generative byproduct.** Note that the learned drifts $v_\theta(\cdot, 1)$ and $v_\theta(\cdot, 0)$ define the distributions $\pi_{\theta,\text{joint}}(x_1|x_0) \approx \pi(x_1|x_0)$ and $\pi_{\theta,\text{ind}}(x_1|x_0) \approx \pi(x_1)$ as solutions to the corresponding SDEs (4). One can simulate these SDEs by utilizing diffusion based generative modeling methods. Samples from $\pi_{\theta,\text{joint}}(x_1|x_0)$ and $\pi_{\theta,\text{ind}}(x_1|x_0)$ for the image based benchmark (Section 5.2) can be seen at Figure 5. This is unnecessary for the MI estimation, but can be considered as an additional feature.

## 5 EXPERIMENTS

We test our method on a diverse set of benchmarks with known ground truth values of mutual information. To cover low-dimensional cases and some basic cases of data lying on a manifold, we employ the tests by (Czyż et al., 2023). Benchmarks from (Butakov et al., 2024b;a) are used to assess the method on manifolds represented as images. To evaluate our method on real-world data, we use protein language model embeddings and adopt the benchmark construction procedure proposed in (Lee & Rhee, 2024). Finally, we evaluate our method on high mutual information tasks from (Czyż et al., 2023). In addition, we present an ablation study in Appendix C.2, analyzing the impact of key hyperparameters, including the volatility coefficient $\epsilon$, neural network architectures. We compare our method with a diverse collection of other MI estimation methods: diffusion based MINDE (Franzese et al., 2024), flow-based NVF, JVF (Dahlke & Pacheco, 2025), MIENF (Butakov et al., 2024a), classical MINE (Belghazi et al., 2018), InfoNCE (van den Oord et al., 2019), fDIME (Letizia et al., 2024), NWJ (Nguyen et al., 2010), KSG (Kraskov et al., 2004).

Across all four experimental setups, our method demonstrates consistently strong performance. Its main advantage emerges in challenging settings involving high-dimensional data and large mutual information values. In these regimes, our approach consistently outperforms both the diffusion-based MINDE estimator and classical MI estimation baselines.

### 5.1 LOW-DIMENSIONAL BENCHMARK

The tests from (Czyż et al., 2023) focus on low-dimensional distributions with tractable mutual information. Various mappings are also applied to make the distributions light- or heavy-tailed, or to non-linearly embed the data into an ambient space of higher dimensionality.

*Info*Bridge is tested with $\epsilon = 1$ and a multi-layer dense neural network is used to approximate the drifts. Our computational complexity is comparable to MINDE (Franzese et al., 2024). For more details, please, refer to Appendix D. In each test, we perform 10 independent runs with 100k train set samples and 10k test set samples, where we train *Info*Bridge neural networks for 100k grad steps. We report the mean MI estimation in Tables 1 and 21.

Overall, the performance of our estimator is similar to that of MINDE (§3) and superior to the classical and flow based, methods, with the Cauchy distribution being the only notable exception. Unfortunately, the Cauchy distribution lacks the first moment, which poses theoretical limitations for Bridge Matching (Shi et al., 2023, Appendix C). However, we overcome this limitation using the tail-shortening asinh transform, which enables our method to achieve near-accurate MI estimates, see first two Asinh columns in Tables 1 and 21.

### 5.2 IMAGE DATA BENCHMARK

In (Butakov et al., 2024b), it was proposed to map low-dimensional distributions with tractable MI into manifolds admitting image-like structure, thus producing synthetic images (in particular, images of 2D Gaussians and Rectangles, see Figures 5a and 5b). By using smooth injective mappings, one ensures that MI is not altered by the transform (Butakov et al., 2024a, Theorem 2.1). In the original works, it is argued that such benchmarks are closer to real data, and therefore closer to realistic setups.

Each neural algorithm is trained with 100k train set samples and validated using 10k samples. In addition we explore experimental setups with less train set samples in Appendix C.3. *Info*Bridge is tested with $\epsilon = 1$, we use a neural network with U-net architecture to approximate the drift and train the model for 100k grad steps with batch size of 64. Other experimental details are reported in Appendix D.3 and the ablation study on neural network architecture and volatility coefficient $\epsilon$ is presented in Appendix C.2.

Table 2: Mean Absolute Error (MAE) averaged for all setups and random seeds, and standard deviations averaged over setups for Image based benchmark. The best result is **bolded**.

| Method | *Info*Bridge | InfoNCE | KSG | MINE | MIENF | NWJ | MINDE-C | MINDE-J |
|---|---|---|---|---|---|---|---|---|
| MAE ↓ | **0.38** | 1.44 | 1.15 | 0.92 | 0.45 | 1.24 | 0.56 | 1.66 |
| Average std. ↓ | 0.07 | 0.04 | 0.02 | 0.13 | 0.08 | 0.08 | 0.43 | 0.45 |

We present our results for $16 \times 16$ and $32 \times 32$ resolution images with both Gaussian and Rectangle structure in Figure 1 and report aggregated MAE and std in Table 2, while the samples from the learned conditional bridge matching models can be viewed in Figures 5c to 5f.

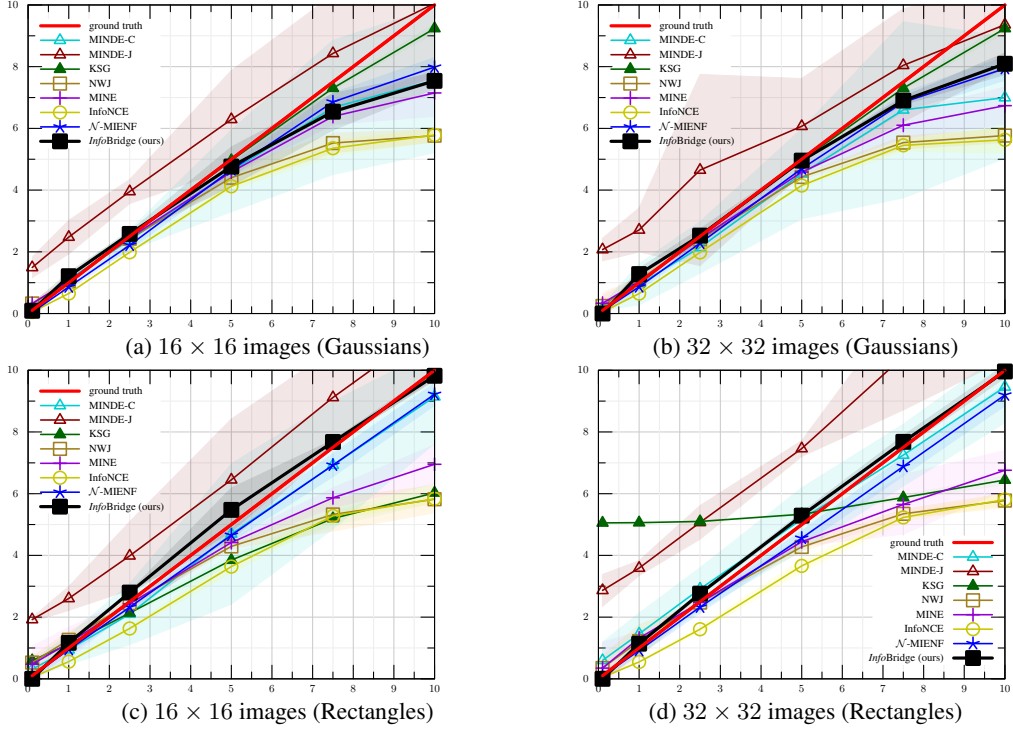

(a) $16 \times 16$ images (Gaussians)

(b) $32 \times 32$ images (Gaussians)

(c) $16 \times 16$ images (Rectangles)

(d) $32 \times 32$ images (Rectangles)

Figure 1: Comparison of the MI estimators. Along $x$ axes is $I(X_0; X_1)$, along $y$ axes is MI estimate $\hat{I}(X_0; X_1)$. We plot 99% confidence intervals acquired from different seed runs.

Our estimator is competitive, being consistently more precise and more stable than both MINDE-C and MINDE-J, and overall slightly better than the previous best-performing method: MIENF (Butakov et al., 2024a). In addition, our method delivers less variance in MI estimation see Table 2.

### 5.3 PROTEIN EMBEDDINGS DATA

We evaluate *Info*Bridge on real-world data, i.e., conduct experiments on protein language model embeddings. Following the benchmark construction method from (Lee & Rhee, 2024), we generate paired datasets with known MI by ensuring that only class labels are shared between variables $X$ and $Y$, making the MI analytically tractable (Lee & Rhee, 2024, §4.5). We use sequence embeddings from the ProtTrans5 model (Elnaggar et al., 2021), based on proteins from A. thaliana and H. sapiens, to create datasets with varying ground truth MI. The final number of training pairs is 20641 and the dimensionality of embed-

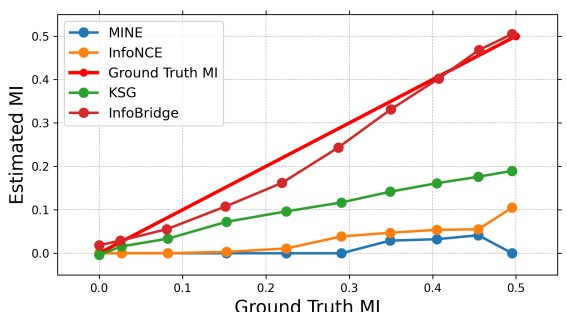

Figure 2: Comparison of the selected estimators on the ProtTrans5 data. Along $x$ axes is ground truth $I(X_0, X_1)$, along $y$ axes is MI estimate $\hat{I}(X_0, X_1)$.

dings is 1024. In addition, the ablation study on the volatility coefficient $\epsilon$ is presented in C.2.

*Info*Bridge with $\epsilon = 1$ is trained for 100 epochs with batch size of 128 and neural network along the other hyperparameters are similar to the low-dimensional benchmark (§5.1) hyperparameters. We compare against MINE, InfoNCE, KSG, and MINDE. All the technical details can be found in Appendix D.4. We present the results plot in Figure 2 and MAE in Table 14. One can see that *Info*Bridge is the only method to consistently estimate MI accurately; MINDE-C and MINDE-J are omitted as they drastically overestimate MI (e.g., MAE: MINDE-C 9.29 vs. *Info*Bridge 0.04).

### 5.4 HIGH MUTUAL INFORMATION

'To evaluate the performance of our method in high mutual information regimes, we conduct experiments on high-dimensional tasks introduced in (Czyż et al., 2023), including correlated uniform, smoothed uniform, correlated gaussian its half cube transformed variant. We systematically vary

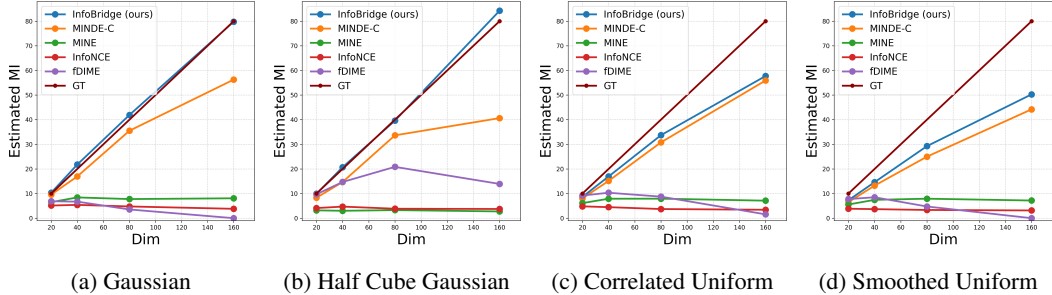

(a) Gaussian     (b) Half Cube Gaussian     (c) Correlated Uniform     (d) Smoothed Uniform

Figure 3: Comparison of MI estimates across dimensions and MI for high mutual information.

the dimensionality and mutual information of the random variables, setting $d \in \{20, 40, 80, 160\}$ and MI $\in \{10, 20, 40, 80\}$. Each experiment uses a training set of 100k samples and a test set of 10k samples. We evaluate our method with $\epsilon = 0.01$ and show the results in Figure 3. The *Info*Bridge is trained for 200 epochs with a batch size of 128, and the neural network is the same as for low-dimensional benchmark § 5.1. To keep the narrative concise, we provide additional baselines and experimental details, refer to Appendix D.5.

As shown, our method yields the most accurate estimates compared to the ground truth, whereas MINE, InfoNCE, and fDIME (GAN, J) fail to capture high mutual information values, and MINDE-C consistently underperforms relative to our approach.

## 6 DISCUSSION

**Potential Impact.** Our contributions include the development of novel unbiased estimator for the MI grounded in diffusion bridge matching theory. The proposed algorithm, *Info*Bridge, demonstrates superior performance compared to commonly used MI estimators on both synthetic and real-world challenging high-dimensional benchmarks. Also, our approach can be used to estimate the KL divergence and differential entropy Appendices B.1 and B.2.

We believe that our work paves the way for new directions in the estimation of MI *in high dimensions*. This has potential real-world applications such as text-to-image alignment (Wang et al., 2024), self-supervised learning (Bachman et al., 2019), deep neural network analysis (Butakov et al., 2024b), and other use cases in high-dimensional settings.

Moreover, our approach offers opportunities for extension by exploring alternative types of bridges within reciprocal processes, for instance, variance-preserving SDE bridges (Zhou et al., 2024), or by incorporating advanced diffusion bridges techniques like time reweighting (Kim et al., 2025).

## 7 LIMITATIONS

Bridge Matching as a general generative modeling paradigm is limited w.r.t. distributions without the first moment. To be well defined Bridge Matching procedure requires some assumptions to be satisfied, which are described in Appendix A.1. If one of the $\pi(x_0)$ or $\pi(x_1)$ distributions doesn't have a first moment, then these assumptions are not satisfied, and our method is not guaranteed to work. One such case can be seen in Table 1, one-degree-of-freedom Student-t distribution, i.e., *St* (dof=1), also known as the Cauchy distribution, that has no first moment.

**Acknowledgements.** The work was supported by the grant for research centers in the field of AI provided by the Ministry of Economic Development of the Russian Federation in accordance with the agreement 000000C313925P4F0002 and the agreement №139-10-2025-033. We thank Kirill Sokolov for the careful feedback on the proofs.

**LLM Usage.** Large Language Models (LLMs) were used only to assist with rephrasing sentences and improving the clarity of the text.

**Reproducibility statement.** All the technical details that are required to reproduce the work are stated either in main experimental section (§ 5) or Appendix D. Additionally, we provide code for the reproduction of experiments in supplementary. All the proofs for provided theoretical results are provided either in Appendix A or Appendix B.

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

## APPENDIX CONTENTS

## A  PROOFS

### A.1  AUXILIARY RESULTS

The following subsection mostly contains results very similar to the ones presented in (Doob, 1957) and the comprehensive derivations are presented for the reader convenience.

**Setup**  Let $\Omega = C([0,1], \mathbb{R}^D)$, $X_t(\omega) = \omega(t)$, $\mathcal{F}_t = \sigma(X_s : s \leq t)$. Let $\pi \in \mathbb{P}(\mathbb{R}^D \times \mathbb{R}^D)$ with marginals $\pi_0, \pi_1$. Let $P^{x_0}$ be Brownian motion with volatility $\sqrt{\varepsilon}$ started at $x_0$. Its transition density is

$$p_t^\varepsilon(x, y) = (2\pi\varepsilon t)^{-D/2} \exp\Big( - \frac{\|y - x\|^2}{2\varepsilon t} \Big).$$

We fix a regular version of the Brownian bridge kernel $W_{|x_0,x_1}^\varepsilon$ which is measurable.

$$Q_\pi := \int W_{|x_0,x_1}^\varepsilon \, d\pi(x_0, x_1),$$

Let $\mu^{x_0} := \pi(\cdot|x_0)$. For $t \in (0,1)$ define the $h$-functions:

$$h^{x_0}(t,x) := \int_{\mathbb{R}^D} \frac{p_{1-t}^{\varepsilon}(x,y)}{p_1^{\varepsilon}(x_0,y)} \, d\mu^{x_0}(y).$$

Intuitively, $h^{x_0}(t,x)$ is the likelihood factor that reweights the reference Brownian motion started at $x_0$ so that, at final time 1, it has terminal law $\mu^{x_0}$.

**Assumption A.1.** $\int_{\mathbb{R}^D} \|x_1\| \, d\pi_1(x_1) < \infty.$

**Lemma A.2.** *Fix $\delta \in (0, 1/2)$ and $x_0 \in \mathbb{R}^D$. Then:*

1. *$h_{\pi}^{x_0}(t,x) \in (0,\infty)$ for all $(t,x) \in [\delta, 1-\delta] \times \mathbb{R}^D$;*

2. *$h_{\pi}^{x_0} \in C^{1,2}([\delta, 1-\delta] \times \mathbb{R}^D)$;*

3. *on each compact $K \subset \mathbb{R}^D$, $\inf_{[\delta,1-\delta] \times K} h_{\pi}^{x_0} > 0$ and $\nabla_x \log h_{\pi}^{x_0}$ is Lipschitz in $x$ on $[\delta, 1-\delta] \times K$.*

*Proof.* Set

$$K_{t,x,x_0}(y) := \frac{p_{1-t}^{\varepsilon}(x,y)}{p_1^{\varepsilon}(x_0,y)}.$$

A direct computation shows that, for each fixed $t \in (0,1)$,

$$K_{t,x,x_0}(y) = C(t) \exp\left( -\frac{t}{2\varepsilon(1-t)} \|y\|^2 + b(t,x,x_0) \cdot y + c(t,x,x_0) \right),$$

where $C(t) = (1-t)^{-D/2}$ and $b,c$ are affine/quadratic expressions in $x, x_0$ (their exact form is not important). Crucially, the coefficient in front of $\|y\|^2$ is *negative* and bounded away from $0$ when $t \in [\delta, 1-\delta]$. Therefore, for such $t$ the function $y \mapsto K_{t,x,x_0}(y)$ is strictly positive and achieves a finite maximum, hence is bounded.

**(1) Positivity and finiteness.** Since $K_{t,x,x_0}(y) > 0$ and bounded in $y$ for each fixed $(t,x)$ with $t \in [\delta, 1-\delta]$, integrating against the probability measure $\mu^{x_0}$ gives $0 < h_{\pi}^{x_0}(t,x) < \infty$.

**(2) $C^{1,2}$ regularity.** For $|\alpha| \leq 2$, the derivatives $\partial_x^{\alpha} K_{t,x,x_0}(y)$ are polynomials in $y$ times the same exponential factor, hence they are also bounded in $y$ for $t \in [\delta, 1-\delta]$. Thus, on any set $[\delta, 1-\delta] \times \{x\}$ we can differentiate under the integral sign by dominated convergence, obtaining that $h_{\pi}^{x_0}$ is $C^{1,2}$ in $(t,x)$.

**(3) Positive lower bound and Lipschitzness of $\nabla \log h$ on compacts.** Fix a compact $K \subset \mathbb{R}^D$. Continuity of $h^{x_0}$ on the compact set $[\delta, 1-\delta] \times K$ implies it attains its minimum there; by (1) this minimum is positive, so $m := \inf_{[\delta,1-\delta] \times K} h^{x_0} > 0$.

Moreover, by (2) the derivatives $\nabla h$ and $\nabla^2 h$ are continuous, hence bounded on $[\delta, 1-\delta] \times K$. Finally, using $\nabla \log h = (\nabla h)/h$ and the identity

$$\nabla^2 \log h = \frac{\nabla^2 h}{h} - \frac{\nabla h \otimes \nabla h}{h^2},$$

we see that $\nabla^2 \log h$ is bounded on $[\delta, 1-\delta] \times K$. A bounded Hessian implies that $\nabla \log h$ is Lipschitz in $x$ on this set. $\qquad \square$

**Proposition A.3.** *For each $x_0$ and each $t < 1$, the conditional laws*

$$Q_{\pi}^{|x_0} := Q_{\pi}(\cdot|X_0 = x_0),$$

*satisfy, for any $A \in \mathcal{F}_t$,*

$$Q_{\pi}^{|x_0}(A) = \mathbb{E}_{P^{x_0}}\left[ \mathbf{1}_A \, h_{\pi}^{x_0}(t, X_t) \right].$$

*Consequently, on every interval $[\delta, 1-\delta]$ the process under $Q_{\pi}^{|x_0}$ solves the SDE (weakly)*

$$dX_t = v_{\pi}^{x_0}(t, X_t) \, dt + \sqrt{\varepsilon} \, dW_t, \qquad v_{\pi}^{x_0}(t,x) := \varepsilon \nabla_x \log h_{\pi}^{x_0}(t,x),$$

*and $v_{\pi}^{x_0}$ is locally Lipschitz in $x$ there.*

*Proof.* Let $A \in \mathcal{F}_t$. By the Markov property of Brownian motion,

$$\mathbb{P}^{x_0}(A, \, X_1 \in dy) = \mathbb{E}_{P^{x_0}}\big[\mathbf{1}_A \, p_{1-t}^\varepsilon(X_t, y)\big] \, dy,$$

and also $\mathbb{P}^{x_0}(X_1 \in dy) = p_1^\varepsilon(x_0, y) \, dy$. Therefore, by Bayes' rule (i.e. by the definition of a regular conditional distribution),

$$W_{|x_0, y}^\varepsilon(A) = \mathbb{P}^{x_0}(A \mid X_1 = y) = \frac{\mathbb{E}_{P^{x_0}}\big[\mathbf{1}_A \, p_{1-t}^\varepsilon(X_t, y)\big]}{p_1^\varepsilon(x_0, y)}.$$

Now integrate the previous identity in $y$ against $\mu_\pi^{x_0}(dy)$. Since $A \in \mathcal{F}_t$, we can pull $\mathbf{1}_A$ outside the $y$-integral:

$$Q_\pi^{|x_0}(A) = \int W_{|x_0, y}^\varepsilon(A) \, \mu_\pi^{x_0}(dy) =$$

$$= \mathbb{E}_{P^{x_0}}\Big[\mathbf{1}_A \int \frac{p_{1-t}^\varepsilon(X_t, y)}{p_1^\varepsilon(x_0, y)} \, \mu_\pi^{x_0}(dy)\Big] = \mathbb{E}_{P^{x_0}}[\mathbf{1}_A \, h_\pi^{x_0}(t, X_t)].$$

In other words, on $\mathcal{F}_t$ the measure $Q_\pi^{|x_0}$ is absolutely continuous w.r.t. $P^{x_0}$ with Radon–Nikodym derivative $h_\pi^{x_0}(t, X_t)$ (Doob, 1957; Léonard, 2014a).

On $[\delta, 1 - \delta]$ Lemma A.2 gives that $h_\pi^{x_0}$ is strictly positive and $C^{1,2}$. Moreover, $x \mapsto p_{1-t}^\varepsilon(x, y)$ solves the heat equation, hence $(t, x) \mapsto h_\pi^{x_0}(t, x)$ solves the backward heat equation on $t < 1$.

Let $L := \frac{\varepsilon}{2}\Delta$ be generator under $P^{x_0}$. Since $(\partial_t + L)h_\pi^{x_0} = 0$ on $t < 1$, Itô's formula gives, for $t \le 1 - \delta$,

$$dh_\pi^{x_0}(t, X_t) = \nabla h_\pi^{x_0}(t, X_t) \cdot \sqrt{\varepsilon} \, dW_t = h_\pi^{x_0}(t, X_t) \, \sqrt{\varepsilon} \, \nabla \log h_\pi^{x_0}(t, X_t) \cdot dW_t.$$

Hence $M_t := h_\pi^{x_0}(t, X_t)$ is a positive $P^{x_0}$-martingale and $dM_t = M_t \, \theta_t \cdot dW_t$ with $\theta_t := \sqrt{\varepsilon} \, \nabla \log h_\pi^{x_0}(t, X_t)$. Define $Q$ on $\mathcal{F}_t$ by $dQ|_{\mathcal{F}_t} = M_t \, dP^{x_0}|_{\mathcal{F}_t}$; then by Girsanov,

$$W_t^Q := W_t - \int_0^t \theta_s \, ds$$

is a Brownian motion under $Q$, and therefore

$$dX_t = \sqrt{\varepsilon} \, dW_t = \varepsilon \nabla \log h_\pi^{x_0}(t, X_t) \, dt + \sqrt{\varepsilon} \, dW_t^Q.$$

Consequently the generator under $Q$ is $L^Q f = \frac{\varepsilon}{2}\Delta f + \varepsilon \nabla \log h_\pi^{x_0} \cdot \nabla f$, i.e. the claimed SDE with drift $v_\pi^{x_0} = \varepsilon \nabla \log h_\pi^{x_0}$ (Léonard, 2013; 2014a).

Local Lipschitzness on compacts follows from Lemma A.2. $\qquad\qquad\square$

**Lemma A.4.** *Assume A.1. Then for each fixed $x_0$ and each $t \in (0, 1)$,*

$$v_\pi^{x_0}(t, x) = \mathbb{E}_{Q_\pi^{|x_0}}\Big[\frac{X_1 - x}{1 - t} \,\Big|\, X_t = x\Big].$$

*Proof.* Under $Q_\pi^{|x_0}$ we first draw $X_1 \sim \mu_\pi^{x_0}$ and then, conditional on $X_1 = y$, run the Brownian bridge from $x_0$ to $y$. Hence, for fixed $t < 1$ and $X_t = x$, the conditional distribution of $X_1$ is proportional to

$$y \;\mapsto\; \frac{p_{1-t}^\varepsilon(x, y)}{p_1^\varepsilon(x_0, y)} \, d\mu_\pi^{x_0}(y).$$

Assumption A.1 implies $\int \|y\| \, d\pi_1(y) < \infty$, and therefore $\mathbb{E}\|X_1\| < \infty$ under $\pi_1$. For $\pi$, note that $\mathbb{E}_\pi \|X_1\|^2 = \mathbb{E}_{\pi_1}\|y\|^2 < \infty$, hence $\mathbb{E}[\|X_1\|^2 | X_0] < \infty$ $\pi_0$-a.s., so $\mu_\pi^{x_0}$ has finite second moment for $\pi_0$-a.e. $x_0$ and the conditional mean is finite.

Differentiate

$$h_\pi^{x_0}(t, x) = \int \frac{p_{1-t}^\varepsilon(x, y)}{p_1^\varepsilon(x_0, y)} \, d\mu_\pi^{x_0}(y)$$

with respect to $x$ (justified for any fixed $t > 0$ exactly as in Lemma A.2). Using the Gaussian identity

$$\varepsilon \nabla_x p_{1-t}^\varepsilon(x, y) = \frac{y - x}{1 - t} \, p_{1-t}^\varepsilon(x, y),$$

we obtain

$$\varepsilon \nabla_x h_\pi^{x_0}(t, x) = \int \frac{y - x}{1 - t} \, \frac{p_{1-t}^\varepsilon(x, y)}{p_1^\varepsilon(x_0, y)} \, d\mu_\pi^{x_0}(y).$$

Now divide both sides by $h_\pi^{x_0}(t, x)$. Since the conditional law of $X_1$ given $X_t = x$ has density proportional to the same integrand, the right-hand side is exactly the conditional expectation of $(X_1 - x)/(1 - t)$. Therefore,

$$v_\pi^{x_0}(t, x) = \varepsilon \nabla_x \log h_\pi^{x_0}(t, x) = \mathbb{E}_{Q_\pi^{|x_0}} \left[ \frac{X_1 - x}{1 - t} \,\middle|\, X_t = x \right].$$

$\square$

## A.2 PROOF OF THEOREM 4.1

*Proof.* Using the disintegration theorem and additive property of relative entropy (Léonard, 2014b, Theorems 1.6, 2.4) at time $t = 0$, we get:

$$\mathrm{KL}\left(Q_\pi \middle\| Q_\pi^{\mathrm{ind}}\right) = \mathrm{KL}\left(\pi(x_0) \middle\| \pi(x_0)\right) + \mathbb{E}_{\pi(x_0)}\left[\mathrm{KL}\left(Q_{\pi|x_0} \middle\| Q_{\pi|x_0}^{\mathrm{ind}}\right)\right].$$

Note that since $Q_\pi, Q_\pi^{\mathrm{ind}}$ share the same marginals at time $t = 0$, first KL term vanishes. In addition, $\mathrm{KL}\left(Q_{\pi|x_0} \middle\| Q_{\pi|x_0}^{\mathrm{ind}}\right) < \infty$ for $\pi_0$-a.e. $x_0$, since the $I(X_0; X_1) < \infty$. Similarly, by using the disintegration theorem again for both times $t = 0, 1$, we get:

$$\mathrm{KL}\left(Q_\pi \middle\| Q_\pi^{\mathrm{ind}}\right) = \mathrm{KL}\left(\pi(x_0, x_1) \middle\| \pi(x_0)\pi(x_1)\right) + \mathbb{E}_{\pi(x_0, x_1)}\left[\mathrm{KL}\left(Q_{\pi|x_0, x_1} \middle\| Q_{\pi|x_0, x_1}^{\mathrm{ind}}\right)\right].$$

Recap that $Q_\pi$ and $Q_\pi^{\mathrm{ind}}$ are both mixtures of Brownian Bridges. Therefore, $Q_{\pi|x_0, x_1} = Q_{\pi|x_0, x_1}^{\mathrm{ind}} = W_{|x_0, x_1}^\epsilon$ and $\mathrm{KL}\left(Q_{\pi|x_0, x_1} \middle\| Q_{\pi|x_0, x_1}^{\mathrm{ind}}\right) = 0$. Then the following holds:

$$\mathrm{KL}\left(Q_\pi \middle\| Q_\pi^{\mathrm{ind}}\right) = \mathrm{KL}\left(\pi(x_0, x_1) \middle\| \pi(x_0)\pi(x_1)\right) = \mathbb{E}_{\pi(x_0)}\left[\mathrm{KL}\left(Q_{\pi|x_0} \middle\| Q_{\pi|x_0}^{\mathrm{ind}}\right)\right]. \tag{16}$$

Moreover, processes $Q_{\pi|x_0}$ and $Q_{\pi|x_0}^{\mathrm{ind}}$ are diffusion processes (§2). Then we can apply the Girsanov Theorem from (Léonard, 2012):

$$\mathrm{KL}\left(Q_{\pi|x_0} \middle\| Q_{\pi|x_0}^{\mathrm{ind}}\right) = \frac{1}{2\varepsilon} \mathbb{E}_{Q_J^{|x_0}} \int_0^{1-\delta} \|v_{\mathrm{joint}}^{x_0}(t, X_t) - v_{\mathrm{ind}}^{x_0}(t, X_t)\|^2 \, dt \tag{17}$$

Now let $\delta \downarrow 0$. The integrand is nonnegative, so the integrals increase with $\delta \downarrow 0$, and by monotone convergence we obtain the same identity with $\int_0^1$ and combine it with (16):

$$\mathrm{KL}\left(\pi(x_0, x_1) \middle\| \pi(x_0)\pi(x_1)\right) = \mathbb{E}_{\pi(x_0)}\left[\mathrm{KL}\left(Q_{\pi|x_0} \middle\| Q_{\pi|x_0}^{\mathrm{ind}}\right)\right] = \tag{18}$$

$$\frac{1}{2\epsilon} \int_0^1 \mathbb{E}_{q_\pi(x_t, x_0)}\left[\|v_{\mathrm{joint}}(x_t, t, x_0) - v_{\mathrm{ind}}(x_t, t, x_0)\|^2\right] dt = I(X_0; X_1),$$

where drifts $v_{\mathrm{joint}}$ and $v_{\mathrm{ind}}$ are defined as in (14) and (15) respectively.

$\square$

## B  EXTENSIONS OF OUR METHODOLOGY

In this appendix section we present:

- Appendix B.1. General KL divergence estimation method with proof and discussion.
- Appendix B.2. Differential entropy estimation result with two possible algorithms and discussion.
- Appendix B.3. Experimental illustrations to the entropy estimation algorithm.
- Appendix B.4. Interaction information estimation method.

### B.1  KL DIVERGENCE ESTIMATOR

In this section, we present a general result for the unbiased estimation of KL divergence between any two distributions $\pi_1(x), \pi_2(x) \in \mathcal{P}(\mathbb{R}^d)$ through the difference of drifts in the SDE formulation (4) of reciprocal processes induced by these distributions, i.e.:

$$Q_{\pi_1} \stackrel{\text{def}}{=} \int W^\epsilon_{|x_0, x_1} d\pi_1(x_1) dp(x_0), \tag{19}$$

$$Q_{\pi_2} \stackrel{\text{def}}{=} \int W^\epsilon_{|x_0, x_1} d\pi_2(x_1) dp(x_0). \tag{20}$$

**Theorem B.1** (KL divergence decomposition). *Consider distributions* $\pi_1(x_1), \pi_2(x_1), p(x_0) \in \mathcal{P}(\mathbb{R}^d)$, *that satisfy some regularity assumptions (Shi et al., 2023, Appendix C), and reciprocal processes* $Q_{\pi_1}, Q_{\pi_2}$ *induced by distributions* $\pi_1(x_1), \pi_2(x_1)$ *(19), (20). Then the KL divergence between distributions* $\pi_1(x_1)$ *and* $\pi_2(x_1)$ *can be represented in the following way:*

$$\text{KL}\left(\pi_1(x_1)\|\pi_2(x_1)\right) = \frac{1}{2\epsilon} \int_0^1 \mathbb{E}_{q_{\pi_1}(x_t, x_0)}\left[\|v^{\pi_1}(x_t, t, x_0) - v^{\pi_2}(x_t, t, x_0)\|^2\right] dt, \tag{21}$$

*where*

$$v^{\pi_1}(x_t, t, x_0) = \mathbb{E}_{q_{\pi_1}(x_1|x_t, x_0)}\left[\frac{x_1 - x_t}{1 - t}\right], \tag{22}$$

$$v^{\pi_2}(x_t, t, x_0) = \mathbb{E}_{q_{\pi_2}(x_1|x_t, x_0)}\left[\frac{x_1 - x_t}{1 - t}\right] \tag{23}$$

*are the drifts of reciprocal processes* $Q_{\pi_1}$ *and* $Q_{\pi_2}$ *respectively. Holds for any distribution* $p(x_0)$.

Our theorem allows us to estimate $\text{KL}\left(\pi_1(x)\|\pi_2(x)\right)$ knowing only the drifts $v^{\pi_1}(x_t, t, x_0)$ and $v^{\pi_2}(x_t, t, x_0)$, which can be recovered using conditional bridge matching (§2). Note that the expression (21) is very similar to the expression (13) in (§4.1). Distribution $p(x_0)$, can be considered as part of the design space and optimised for each particular problem.

The KL divergence is a fundamental quantity, and its estimator can have many applications, such as mutual information estimation or entropy estimation using results described in B.2.

*Proof.* Consider

$$\text{KL}\left(\pi_1(x_1)p(x_0)\|\pi_2(x_1)p(x_0)\right) = \underbrace{\text{KL}\left(p(x_0)\|p(x_0)\right)}_{=0} + \mathbb{E}_{p(x_0)}\left[\text{KL}\left(\pi_1(x_1|x_0)\|\pi_2(x_1|x_0)\right)\right] =$$

$$= \mathbb{E}_{p(x_0)}\left[\text{KL}\left(\pi_1(x_1)\|\pi_2(x_1)\right)\right] = \text{KL}\left(\pi_1(x_1)\|\pi_2(x_1)\right),$$

Next to get

$$\text{KL}\left(\pi_1(x_1)p(x_0)\|\pi_2(x_1)p(x_0)\right) = \frac{1}{2\epsilon}\int_0^1 \mathbb{E}_{q_{\pi_1}(x_t, x_0)}\left[\|v^{\pi_1}(x_t, t, x_0) - v^{\pi_2}(x_t, t, x_0)\|^2 dt\right],$$

one can repeat all the steps that were taken to show:

$$\mathrm{KL}\left(\pi(x_0, x_1) \| \pi(x_0)\pi(x_1)\right) = \frac{1}{2\epsilon} \int_0^1 \mathbb{E}_{q_\pi(x_t, x_0)}\left[\|v_{\text{joint}}(x_t, t, x_0) - v_{\text{ind}}(x_t, t, x_0)\|^2\right] dt,$$

in the proof of Theorem 4.1.

$\square$

## B.2 DIFFERENTIAL ENTROPY ESTIMATOR

A general result on the information projections and maximum-entropy distributions suggests a way of calculating differential entropy through the KL divergence estimation.

**Theorem B.2** (Theorem 6.7 in (Kappen, 2024)). *Let $\phi\colon \mathbb{R}^n \to \mathbb{R}^k$ be any measurable function, an absolutely continuous probability distribution $p(x) \in \mathcal{P}(\mathbb{R}^d)$ and define $\alpha \stackrel{def}{=} \mathbb{E}_p \phi(x)$. Now, for any $\theta \in \mathbb{R}^k$ consider an absolutely continuous probability distribution $q_\theta \in \mathcal{P}(\mathbb{R}^d)$ with density:*

$$q_\theta(x) = \exp(\langle\theta, \phi(x)\rangle - A(\theta)), \quad A(\theta) = \log \mathbb{E}_{q_\theta(x)} e^{\langle\theta, \phi(x)\rangle}.$$

*If there exists $\theta^*$ such that $p$ is absolutely continuous w.r.t. $q_{\theta^*}$ and $\mathbb{E}_{q_{\theta^*}(x)} \phi(x) = \alpha$, then*

$$H(p) = H(q_{\theta^*}) - \mathrm{KL}\left(p\|q_{\theta^*}\right),$$

**Corollary B.3.** *Let $X$ be a $d$-dimensional absolutely continuous random vector with probability density function $p$, mean $m$ and covariance matrix $\Sigma$. Then*

$$H(p) = H\left(\mathcal{N}(m, \Sigma)\right) - \mathrm{KL}\left(p\|\mathcal{N}(m, \Sigma)\right), \qquad H\left(\mathcal{N}(m, \Sigma)\right) = \frac{1}{2} \log\left((2\pi e)^d \det \Sigma\right),$$

*where $\mathcal{N}(m, \Sigma)$ is a Gaussian distribution of mean $m$ and covariance matrix $\Sigma$.*

**Corollary B.4.** *Let $X$ be an absolutely continuous random vector with probability density function $p$ and $\mathrm{supp}\, X = S$, where $S$ has finite and non-zero Lebesgue measure $\mu(S)$. Then*

$$H(p) = H\left(\mathrm{U}(S)\right) - \mathrm{KL}\left(p\|\mathrm{U}(S)\right), \qquad H\left(\mathrm{U}(S)\right) = \log \mu(S),$$

*where $\mathrm{U}(S)$ is a uniform distribution on $S$.*

Similar results can also be obtained for other members of the exponential family. The first result (B.3) can be considered as a general recipe, while the second one (B.4) can be useful when we have prior knowledge about the support of $X$ being restricted. Approach described in B.2 is very flexible and can be considered as a generalization of the method used in (Franzese et al., 2024).

In practice, to estimate the entropy of some probability distribution, it is sufficient to follow one of the described in B.4 or B.3. For example, if one uses Corollary B.3: 1) estimate mean $m$ and covariance matrix $\Sigma$ using a set of data samples, 2) calculate entropy $H(\mathcal{N}(m, \Sigma))$ via the closed form expression, 3) calculate the KL divergence $\mathrm{KL}\left(p\|\mathcal{N}(m, \Sigma)\right)$ via learning two conditional diffusion bridges models and utilizing our estimator (21) from B.1.

## B.3 ENTROPY ESTIMATION EXPERIMENTS

We conduct entropy estimation experiments using our method proposed in Appendix B.2. We estimate the entropy of high-dimensional ($D = 10$) exponential and uniform distributions $\pi$ with varying distribution parameters. Corollary B.3 (Gaussian) method is used for the both distributions and Corollary B.4 (Uniform) method for the uniform distribution, since is has finite support. For each of the setups we prepare the dataset of 8000 training and 2000 test samples. Additionally we explore different reference distributions $p(x_0)$, see Theorem B.1, for the $\mathrm{KL}\left(\cdot\|\mathcal{N}(m, \Sigma)\right)$ or $\mathrm{KL}\left(\cdot\|U(S)\right)$ estimation. We explored four reference distributions $p(x_0)$: low ($\epsilon = 0.1$), medium ($\epsilon = 1$), and high ($\epsilon = 10$) variance Gaussians (all with zero mean), and a "data Gaussian" with mean and standard deviation derived from the training data.

One can observe that *Info*Bridge is capable of entropy estimation on practice. Versions with low/medium variance gaussians generally show superior performance to high variance gaussian and data gaussian. In the case of uniform distribution (Corollary B.4) method performs a lot better, suggesting that it is preferable in the cases when it can be applied (restricted support distributions).

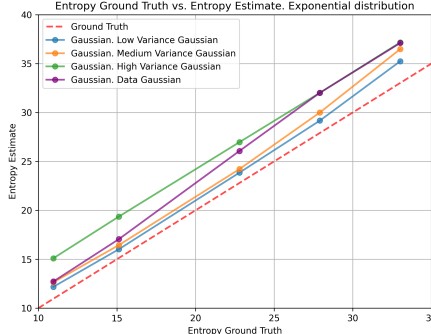 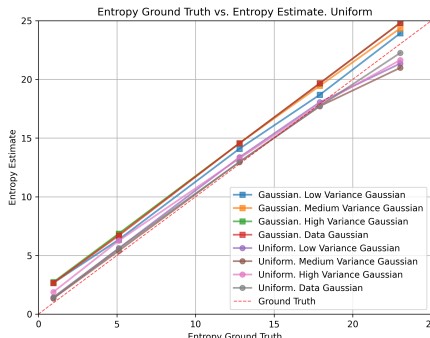

Figure 4: Comparison of our *Info*Bridge combined with different methods Corollary B.3 (Gaussian) and Corollary B.4 (Uniform) and with different reference distributions $p(x_0)$ on the entropy estimation task. Along $x$ axes is the ground truth entropy $H(X)$, along $y$ axes is the entropy estimate $\hat{H}(X)$. Each configuration is ran once.

## B.4 INTERACTION INFORMATION

Here we propose the generalization of *Info*Bridge for the *interaction information* estimation, which is the generalization of mutual information for more than two random variables (Rosas et al., 2019). Interaction information for random variables $X_0, X_1, X_2$ is defined by:

$$I(X_0; X_1; X_2) \stackrel{\text{def}}{=} I(X_0; X_1) - I(X_0; X_1|X_2) =$$
$$= \text{KL}\left(\Pi_{X_0,X_1} \| \Pi_{X_0} \otimes \Pi_{X_1}\right) - \text{KL}\left(\Pi_{X_0,X_1|X_2} \| \Pi_{X_0|X_2} \otimes \Pi_{X_1|X_2}\right). \tag{24}$$

This definition can be generalized for more random variables in a similar way. Both MI information terms can be estimated using Theorem 4.1 and our practical algorithm *Info*Bridge. Applications of interaction information include neuroscience (Bounoua et al., 2024b), climate models (Runge et al., 2019), econometrics (Dosi & Roventini, 2019).

## B.5 MUTUAL INFORMATION BETWEEN RANDOM VARIABLES OF DIFFERENT DIMENSIONALITY

Here we outline the ways to estimate the mutual information between variables of different dimension. In turn this allows to calculate the mutual information between random variables from different modalities, e.g., text and images.

- **Zero or noise padding.** One can embed the lower dimensional variable into the higher dimensional space by padding the spare dimensions with zeros or noise. Such padding does not affect the mutual information, as (1) the padding can be decoupled from the original random variable, and (2) it remains uncorrelated with the other random variables.

- **Employ the general KL estimation result.** One can employ our general KL estimation result, i.e., Theorem B.1. There, we propose to add a third distribution, e.g., a point, normal distribution, or distribution of images, to start our two diffusion bridge matching models from. One can set $\pi_1(x) = q(x_1, x_2)$ and $\pi_2(x) = q_1(x_1)q_2(x_2)$, where $x_1$ and $x_2$ random variables of different dimension, i.e., $x_1 \in \mathbb{R}^{d_1}$ and $x_2 \in \mathbb{R}^{d_2}$.

- **Autoencoder-based latent alignment.** Another approach is to embed both random variables into a common latent space via autoencoders. Since a properly trained autoencoder preserves mutual information (Butakov et al., 2024b), MI estimation can then be carried out in this shared latent space.

## C  ADDITIONAL DISCUSSION ON METHOD

### C.1  CONCEPTUAL COMPARISON WITH MINDE

While MINDE also utilizes diffusion models for the MI estimation task, our method is built with totally different diffusion processes in mind. Utilization of regular diffusion generative modeling processes in MINDE has several disadvantages that our method particularly addresses.

**First**, by leveraging finite-time bridge processes instead of infinite-time diffusion (which in practice is approximated with a finite-time horizon), we provide an **unbiased** estimator. In particular, one can see that the MINDE MI estimator (9) has an intractable bias term in it, i.e., $\mathrm{KL}(q_T^A \| q_T^B)$, while our MI estimator (13) is exact.

**Second**, MINDE treats MI estimation as a **generative modeling task**, learning drifts to transform $\mathcal{N}(0, I)$ into $\Pi_{X_1}$ or $\Pi_{X_1|X_0}$. In contrast, our method frames it it as a **domain transfer task**, directly transforming input marginal $\Pi_{X_0}$ to distributions $\Pi_{X_1|X_0}$ or $\Pi_{X_1}$. This lead to two consequences:

- In domain transfer the diffusion bridge process is *easier to learn*, since it both starts and ends at data. This is a contrast to complex generation diffusion process that starts at noise and ends at data. The diffusion bridge approaches for domain transfer with diffusion process starting at data are known to be superior to conditional diffusion models (Liu et al., 2023; Zhou et al., 2024).

- In our method each of the diffusion process starting at some particular point $x_0$ is a Schrödinger Föllmer process (Vargas et al., 2023), which is a special instance of Schrodinger Bridge (Bortoli et al., 2024; Vargas et al., 2021). Trajectories of Schrodinger Bridge processes with small volatility coefficient $\epsilon$ require less energy for translation and are relatively "straight". As a result, these trajectories are easy to follow, are expected to deliver *less numerical error* when dealt with and are easier to learn.

To *empirically support the mentioned superiorities* of our domain translation trajectories over MINDE's data generation trajectories, we compute the variance of MI estimation, which is based on trajectory Monte Carlo integration (Eq (13) for our method, Eq 19 (Franzese et al., 2024) for MINDE). Specifically, we evaluate MINDE-C and our method on the image-based benchmark (§ 5.2) using 16×16 rectangular images. We vary the number of sample tuples $\{x_0^i, x_t^i, x_T^i\}_{i=1}^N$ for MINDE-C and $\{x_0^i, x_t^i, x_1^i\}_{i=1}^N$ for our method and run 10 MI estimates per setting, and report the mean $\pm$ standard deviation in Table 3.

| Method \ N samples | 128 | 1024 | 4096 | 16384 | 65536 | 131072 |
|---|---|---|---|---|---|---|
| InfoBridge (ours) | $9.79 \pm 0.72$ | $9.84 \pm 0.34$ | $9.9 \pm 0.1$ | $9.84 \pm 0.06$ | $9.86 \pm 0.028$ | $9.84 \pm 0.028$ |
| MINDE | $12.20 \pm 6.65$ | $11.46 \pm 3.32$ | $10.62 \pm 3.94$ | $12.59 \pm 2.53$ | $11.17 \pm 1.38$ | $10.67 \pm 0.75$ |

Table 3: Comparison of MI estimation accuracy across different sample sizes.

Our method achieves *1–2 orders of magnitude lower standard deviation* in MI estimates compared to MINDE. Even with $1000\times$ more samples, MINDE only approaches our accuracy. This is due to its reliance on complex noise-to-data trajectories, which make its estimates highly volatile and unreliable, unlike our more stable data-to-data paths. As shown in Figure 1, MINDE consistently exhibits the largest and most unstable confidence intervals, undermining the reliability and precision of its MI estimates. Furthermore, every sample requires inference of a neural network and hence the reliable estimation of MI by MINDE requires several orders of magnitude more computational resources.

### C.2  ABLATION STUDY

Here we vary different important hyperparameters of our *Info*Bridge method.

**Volatility coefficient** $\epsilon$. We evaluate different volatility coefficients on the image benchmark (§5.2) and protein embeddings data (§5.3). We run the image benchmark experiment on Gaussian 16x16 setting with ground truth MI=10 and the full protein embeddings data experiment with one random seed and report mean MAE (Mean Absolute Error) error for the last 5 measurements during training,

in both of setups. Results are presented in Table 4 and Table 5. One can see that MAE relatively stable for all the setups, while the optimal $\epsilon$ values is different for the both experiments.

One can notice that $\epsilon = 1$ works relatively well on both of the setups. Furthermore, $\epsilon = 1$ performs well on all the experiments we have done, we use it for the three out of four experiments in the main part of the paper and show its competitiveness on the high MI experiment in Appendix D.5. In that light we call $\epsilon = 1$ a **robust choice** suitable for all the data.

We consider $\epsilon$ hyperparameter as a tool for **bias/variance trade-off**. The bigger $\epsilon$ hyperparameter corresponds to the reduction of variance and growth of bias, i.e., the learning procedure is very stable but slow, MI is mostly underestimated, and the method is stable w.r.t. other hyperparameters. While a smaller $\epsilon$ hyperparameter corresponds to the growth of variance and reduction in bias, i.e., the learning procedure has more variance but converges faster, and the model is less stable w.r.t. other hyperparameters. The general advice is to take $\epsilon$ as low as possible to keep the learning procedure stable. In most cases, $\epsilon = 1$ is enough to get decent results and keep the model stable.

Table 4: Mean Absolute Error (MAE) values on Image data benchmark (Gaussian 16x16) with ground truth MI=10 experiment for the different $\epsilon$ values. Each configuration (expect for $\epsilon = 1$) is ran one time and 5 last MI measurements during training are averaged to produce the shown result. The best configuration is **bolded** and configuration used in the main part of the paper is underlined.

| $\epsilon$ | 0.001 | 0.003 | 0.01 | 0.03 | 0.1 | 0.3 | 1 | 3 | 10 |
|---|---|---|---|---|---|---|---|---|---|
| MAE $\downarrow$ | 2.09 | 1.79 | **1.53** | **1.53** | 1.72 | 1.79 | 2.46 | 2.82 | 6.45 |

Table 5: Mean Absolute Error (MAE) values on ProtTran5 experiment, averaged over all the ground truth MI values, for the different $\epsilon$ values. Each configuration (expect for $\epsilon = 1$) is ran one time and 5 last MI measurements during training are averaged to produce the shown result. The best configuration is **bolded** and configuration used in the main part of the paper is underlined.

| $\epsilon$ | 0.1 | 0.3 | 1 | 3 | 10 |
|---|---|---|---|---|---|
| MAE $\downarrow$ | 0.775 | 0.232 | **0.04** | 0.089 | 0.129 |

**Number of parameters in a neural network.** The width of the Unet (Ronneberger et al., 2015) neural network architecture used in *Info*Bridge on Image benchmark experiments is varied to yield different number of parameters of a neural network. Numbers of base filters, $256, 64, 32, 16, 8$, yield number of parameters, $27M, 1.7M, 428K, 108k, 28k$. Each configuration, except for 256 number of filters and $27M$ parameters is run once, and the final five mutual information (MI) measurements during training are averaged to obtain the reported results. Outcomes are presented in Table 6. As one can see our *Info*Bridge is robust w.r.t. neural network architecture.

Table 6: Comparison of the *Info*Bridge with different neural network architectures on the Image data benchmark ($16 \times 16$ Gaussians setup) with ground truth MI=7.5. Each configuration is ran one time and 5 last MI measurements during training are averaged to produce the shown result. The configuration used in the main experimental part is underlined.

| N Params | GT MI=7.5 (mean $\pm$ std) |
|---|---|
| 28k | $7.56 \pm 0.3$ |
| 108k | $6.67 \pm 0.15$ |
| 428k | $6.67 \pm 0.13$ |
| 1.7M | $6.52 \pm 0.07$ |
| 27M | $6.59 \pm 0.17$ |

**Training dynamics.** Here we show the training dynamics of our *Info*Bridge on the High MI experiment within the Smoothed Uniform distributions setup from Section 5.4. Originally we train our model for 200 epochs and the Table shows the MI estimation on the different stages of training:

10, 20, 30, 60, 120, 200 epochs. The results are presented in Table 7. One can see that our method is **robust** w.r.t. number of training iterations or number of training epochs used.

| $N$ epochs GT MI | 10 | 20 | 40 | 80 |
|---|---|---|---|---|
| 10 | 8.14 | 16.09 | 30.87 | 47.70 |
| 20 | 7.79 | 15.51 | 30.26 | 48.61 |
| 30 | 7.70 | 15.50 | 30.47 | 50.16 |
| 60 | 7.51 | 15.02 | 29.67 | 50.20 |
| 120 | 7.29 | 14.63 | 29.55 | 50.01 |
| 200 | 7.01 | 14.32 | 28.89 | 50.82 |

Table 7: The MI estimation of ours *Info*Bridge method is shown for different training stages. The experimental setup is High MI test with Smoothed Uniform distributions. The same $10, 20, 40, 80$ MI distributions, as in the main text are used.

### C.3 TRAIN SAMPLE SIZE STUDY

Here we study the stability of our *Info*Bridge method w.r.t. the size of train dataset. While in main text we study datasets that do consist of 100k train samples in all the cases except for protein data experiment Section 5.3, in this section we test lower train samples sizes of 5k and 10k.

In particular, we take the Rectangle $16 \times 16$ experiments setup from Section 5.2, where we do vary the number of training samples: 100k, 10k and 5k. The number of test samples is the same 10k. The results from $100k$ are taken directly from the original experiment in Section 5.2. For MINDE and our *Info*Bridge methods we do make neural networks smaller to prevent overfitting.

| Method | $N$ train samples | $N$ params | MAE | 0.1 | 1 | 2.5 | 5 | 7.5 | 10 |
|---|---|---|---|---|---|---|---|---|---|
| InfoBridge (ours) | 5k | 106k | **0.199** | 0.008 | 0.79 | 2.33 | 4.83 | 7.20 | 9.75 |
| MINDE | 5k | 106k | 2.397 | 0.47 | 0.36 | 0.44 | 0.38 | 4.90 | 5.91 |
| MINE | 5k | 50k | 1.618 | 0.00 | 0.60 | 2.00 | 3.74 | 4.77 | 5.28 |
| InfoBridge (ours) | 10k | 428k | **0.142** | 0.006 | 1.02 | 2.65 | 5.24 | 7.65 | 9.80 |
| MINDE | 10k | 428k | 1.663 | 0.46 | 0.44 | 0.48 | 5.93 | 6.93 | 4.46 |
| MINE | 10k | 50k | 1.513 | 0.00 | 0.78 | 2.10 | 3.90 | 4.85 | 5.39 |
| InfoBridge (ours) | 100k | 27M | **0.228** | 0.00 | 1.16 | 2.79 | 5.47 | 7.67 | 9.82 |
| MINDE | 100k | 27M | 0.815 | 0.41 | 0.91 | 2.45 | 4.22 | 5.63 | 8.21 |
| MINE | 100k | 50k | 0.978 | 0.47 | 1.15 | 2.43 | 4.41 | 5.86 | 6.95 |

Table 8: Image benchmark experiment with Rectangle $16 \times 16$ setup with variable train samples size. MAE stands for Mean Absolute Error. The best method along each $N$ train samples setup is **bolded**.

The results are presented in Table 8. One can see that our method's performance stays almost the same, while MINDE and MINE performance degrade. This shows the robustness of our method w.r.t. the size of train dataset.

### C.4 STRATEGIES FOR DRAWING TRAIN SAMPLES

Here we explore several design choices for sampling the data utilized for computation of the independent drift function loss $\mathcal{L}_\theta^2$ in the Algorithm 1 . In (Letizia et al., 2024) authors argue that **independent permutation** used in the Algorithm 1 introduces additional correlation. In practice, even the simplest one as regular permutation works well, but it can be replaced by either **"derangement"** permutation (Letizia et al., 2024) or **new marginal draws**.

We test these strategies on the High MI experiment with the Smoothed Uniform distribution. All the ours *Info*Bridge method hyperparameters (except for the training data sampling method) match the ones used in the main text. The results are presented in Table 9 and show a **slight performance improvement** for both derangement and fresh marginal draws strategies across all the setups.

| Samples draw strategy | MAE | MI=10 | MI=20 | MI=40 | MI=80 |
|---|---|---|---|---|---|
| Independent Permutation* | 12.24 | 7.01 | 14.32 | 28.89 | 50.82 |
| Derangement | **10.51** | 7.24 | 14.74 | **31.21** | 54.78 |
| Fresh sample draw | 10.62 | **7.31** | **14.76** | 30.06 | **55.38** |

Table 9: Study on methods for drawing independent samples for the computation of $\mathcal{L}_\theta^2$ in Algorithm 1. The experiment setup is High MI (see Section 5.4) with Smoothed Uniform distributions. * - states for method used in the main paper experiments. In the top MI states for ground truth MI and the closer results to the ground truth the better. The best MI estimation is **bolded** for each MI setup.

## C.5 One neural network approach versus two neural networks approach

In this section, we empirically justify the one neural network choice described in Section 4.2. In particular, we take Image Data experiment Rectangle $16 \times 16$ setup from Section 5.2 and try two drift functions parameterizations: 1) **one neural network approach** described in Section 4.2 and 2) **two neural networks approach** where each $v_{\text{joint}}$ and $v_{\text{ind}}$ parametrized by different neural networks. Both approaches have hyperparameters that do match ones used for the experiment in the Section 5.2 fully. The results are presented in Table 10. One can see that the two neural networks approach consistently overestimates the MI and generally provides more error. That is why we stick to the one neural network approach.

| Model | GT MI=0.1 | GT MI=1 | GT MI=2.5 | GT MI=5 | GT MI=7.5 | GT MI=10 |
|---|---|---|---|---|---|---|
| One Neural Network | 0.00 | **1.16** | **2.79** | **5.47** | **7.67** | **9.82** |
| Two Neural Networks | **0.83** | 1.86 | 3.49 | 5.93 | 8.12 | 10.23 |

Table 10: The comparison between the parametrizations for drift functions $v_{\text{joint}}$ and $v_{\text{ind}}$, i.e., one neural network approach and two neural networks approach. The comparison is done on Image Data benchmark Rectangle $16 \times 16$ setup. The table shows MI estimation and GT MI in the top row stands for ground truth MI. The best result along each setup is **bolded**.

## D Experimental supplementary

In this section, additional experimental results and experimental details are described.

### D.1 Metrics

**MAE.** Mean Absolute Error (MAE) is computed as follows:

$$\text{MAE}(x, y) = |x - y|.$$

We measure the MAE between MI estimates and ground truth MI values. In Image Data benchmark analysis Table 2 the MAE is shown averaged over all of the experiments including the different seed runs. In Appendix C.2 Ablation study Table 4 the MAE is averaged over last 5 MI estimations during training and in Table 5 the MAE averaged over all the experimental setups and 5 last MI estimations during training.

**Std.** Standard deviation (std) is computed over samples as follows:

$$\text{std}(\{x_i\}_{i=0}^N) = \sqrt{\frac{1}{N-1}\sum_{i=0}^{N}(x_i - \overline{x})},$$

where $\overline{x} = \frac{1}{N}\sum_{i=0}^{N} x_i$.

Table 11: Neural networks hyperparametes for low-dimensional (Czyż et al., 2023) benchmark. "Dim" - dimensionality of a MI estimation problem, "Filters" – number of filters in MLP, "Time Embed" – number filters in time embedding, "Parameters" – number of overall neural network parameters.

| Dim | Filters | Time Embed | Parameters |
|---|---|---|---|
| $\leq 5$ | 64 | 64 | 43K |
| 25 | 128 | 128 | 176K |
| 50 | 256 | 256 | 699K |

## D.2 LOW-DIMENSIONAL BENCHMARK

**Additional results.** In Table 21 we present the results of low-dimensional benchmark (Czyż et al., 2023) with precision of 0.01 nats.

**Experimental details.** The benchmark implementation is taken from the official repository:

$$\texttt{https://github.com/cbg-ethz/bmi}$$

***Info*Bridge.** Neural networks are taken of almost the same architecture as in (Franzese et al., 2024), which is MLP with residual connections and time embedding. Additional input $s$ described in (§ 4.2) and is processed the same way as time input. Number of parameters is taken depending on a dimensionality of the problem, see Table 11. Exponential Moving Average as a widely recognized training stabilization method was used with decay parameter of 0.999. For all the problems neural networks are trained during 100k iterations with $\epsilon = 1$, batch size 512, lr 0.0003. Mutual Information is estimated by Algorithm 2 with $N$ pairs of samples $\{x_t^i, x_0^i, t^i\}_{i=1}^N$, where $N$ is equal to the number of test samples times 10, i.e., 100k. The time $t$ is sampled uniformly in $t \in [0, 1 - 10^{-3})$ segment. The approximate runtime was under the 20 minutes for each of the experimental setup on NVIDIA A100 GPU.

**NVF and JVF.** The implementation of NVF and JVF is taken from the official repository:

$$\texttt{https://github.com/calebdahlke/FlowMI}$$

and used with all the provided by authors default hyperparameters for the benchmark. The methods are ran on CPU and execution takes less then two hours for each benchmark problem run. The size of a neural network is from 10k to 400k parameters depending of the task dimensionality and the submethod. The approximate runtime was under the 30 minutes under the 30 minutes for NVF and under the 1 hours for JVF for each of the experimental setup on NVIDIA A100 GPU.

## D.3 IMAGE DATA BENCHMARK

**Random seeds.** We run *Info*Bridge, MIENF, and MINDE with 3 random seeds, and all other methods with 5 seeds per experimental setup. In Figure 1 and Table 2, we report the mean performance across seeds, along with standard deviations and confidence intervals computed from the seed-wise variability.

**Experimental details for *Info*Bridge.** The implementation of image data benchmark (Butakov et al., 2024a) was taken from the official repository:

$$\texttt{https://github.com/VanessB/mutinfo}$$

Following authors of the benchmark, *Gaussian* images were generated with all the default settings, *Rectangle* images were also generated with all the default settings, but minimum size of rectangle is 0.2 to avoid singularities. All the covariance matrices for the distributions defining the mutual information in the benchmark were generated without randomization of component-wise mutual information, but with randomization of the off-diagonal blocks of the covariance matrix.

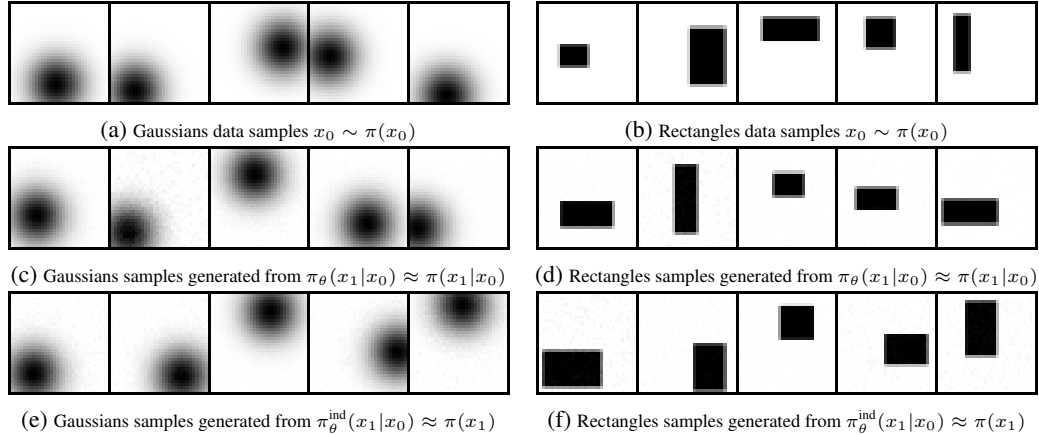

(a) Gaussians data samples $x_0 \sim \pi(x_0)$      (b) Rectangles data samples $x_0 \sim \pi(x_0)$

(c) Gaussians samples generated from $\pi_\theta(x_1|x_0) \approx \pi(x_1|x_0)$    (d) Rectangles samples generated from $\pi_\theta(x_1|x_0) \approx \pi(x_1|x_0)$

(e) Gaussians samples generated from $\pi_\theta^{\mathrm{ind}}(x_1|x_0) \approx \pi(x_1)$    (f) Rectangles samples generated from $\pi_\theta^{\mathrm{ind}}(x_1|x_0) \approx \pi(x_1)$

Figure 5: Examples of synthetic images from the (Butakov et al., 2024a) benchmark can be seen in Figures 5a and 5b. Note that images are high-dimensional, but admit latent structure, which is similar to real datasets. Samples generated by our *Info*Bridge from the learned distributions $\pi_\theta(x_1|x_0) \approx \pi(x_1|x_0)$ and $\pi_\theta^{\mathrm{ind}}(x_1|x_0) \approx \pi(x_1)$ defined as solutions to SDEs (4) with approximated drifts $v_\theta(\cdot, 0)$ and $v_\theta(\cdot, 1)$, respectively, can be seen in Figures 5c to 5f. All the images have $32 \times 32$ resolution.

To approximate the drift coefficient of diffusion U-Net (Ronneberger et al., 2015) with time, conditional neural networks were used, special input $s$ was processed as time input. For all the tests, neural networks were the same and had 2 residual layers per U-net block with 256 base channels, positional timestep encoding, upscale and downscale blocks consisting of two resnet blocks, one with attention and one without attention. The number of parameters is $\sim$27M. During the training, 100k gradient steps were made with batch size of 64 and learning rate 0.0001. Exponential moving average was used with decay rate 0.999. Mutual Information was estimated by Algorithm 2 with $N$ pairs of samples $\{x_t^n, x_0^n, t^n\}_{i=n}^N$, where $N$ is equal to the number of test samples, i.e., 10k. Nvidia A100 was used for the *Info*Bridge training. Each run (one seed) took around 6 and 18 GPU-hours for the $16 \times 16$ and $32 \times 32$ image resolution setups, respectively. The time $t$ is sampled uniformly in $t \in [0, 1 - 10^{-3})$ segment.

**Visualization of learned distribution.** Figure 5 presents samples generated from the learned distributions alongside ground truth samples from the dataset. It is evident that the samples from both $\pi_\theta(x_1|x_0)$ and $\pi_\theta^{\mathrm{ind}}(x_1|x_0)$ closely resemble those drawn from $\pi(x_0)$, indicating that the marginal distribution of $x_0$ is well approximated by the learned models.

**Experimental details for other methods.** In this part, we provide additional experimental details regarding other methods featured in Figure 1. We report the NN architectures used for neural estimators in Table 12.

Table 12: The NN architectures used to conduct the tests with images in 5.

| NN | | Architecture |
|---|---|---|
| GLOW, $16 \times 16$ ($32 \times 32$) images | $\times 1$: | 4 (5) splits, 2 GLOW blocks between splits, $\quad\quad\quad\quad\quad^{\times 2 \text{ in parallel}}$ 16 hidden channels in each block, leaky constant $= 0.01$ |
| | $\times 1$: | Orthogonal projection linear layer$^{\times 2 \text{ in parallel}}$ |
| Critic NN, $16 \times 16$ ($32 \times 32$) images | $\times 1$: | [Conv2d(1, 16, ks=3), MaxPool2d(2), LeakyReLU(0.01)]$^{\times 2 \text{ in parallel}}$ |
| | $\times 1(2)$: | [Conv2d(16, 16, ks=3), MaxPool2d(2), LeakyReLU(0.01)]$^{\times 2 \text{ in parallel}}$ |
| | $\times 1$: | Dense(256, 128), LeakyReLU(0.01) |
| | $\times 1$: | Dense(128, 128), LeakyReLU(0.01) |
| | $\times 1$: | Dense(128, 1) |

**MINDE.** We use the official implementation of MINDE:

`https://github.com/MustaphaBounoua/minde`

As a neural network, we adapt same U-net architecture as for our *Info*Bridge in image data experiments with approximately the same number of parameters , i.e., 27M. Training procedure hyperparameters do match the *Info*Bridge, i.e., 100k gradient steps with batch size 64 and learning rate 0.0001, exponential moving average with 0.999 decay rate. Mutual Information was estimated using 10 samples of $\{t^i, x_t^i\}$ per each pair $\{x_0^i, x_1^i\}$ in the test dataset. Nvidia A100 was used to train the models. Each run (one seed) took around 2 and 5 GPU-hours for the $16\times16$ and $32\times32$ image resolution setups.

**MIENF.** Under MIENF we consider gaussian-based $\mathcal{N}$-MIENF. This method is based on bi-gaussianization of the input data via a Cartesian product of learnable diffeomorphisms (Butakov et al., 2024a). Such approach allows for a closed-form expression to be employed to estimate the MI.

With only minor stability-increasing changes introduced, we adopt the Glow (Kingma & Dhariwal, 2018) flow network architecture from (Butakov et al., 2024a), which is also reported in 12 ("GLOW"). We used the `normflows` package (Stimper et al., 2023) to implement the model. Adam (Kingma & Ba, 2017) optimizer was used to train the network on $10^5$ images with a batch size 512, and the learning rate decreasing from $5 \cdot 10^{-4}$ to $10^{-5}$ geometrically. For averaging, we used 3 different seeds. Nvidia A100 was used to train the flow models. Each run (one seed) took no longer than four GPU-hours to be completed. The size of normalizing flow neural network is around 200k parameters.

**KSG.** Kraskov-Stögbauer-Grassberger (Kraskov et al., 2004) mutual information estimator is a well-known $k$-NN non-parametric method, which is very similar to unweighted Kozachenko-Leonenko estimator (Kozachenko & Leonenko, 1987). This method employs distances to $k$-th nearest neighbors to approximate the pointwise mutual information, which is then averaged.

We used $k = 1$ (one nearest neighbour) for all the tests. The number of samples was $10^5$ for Gaussian images and $10^4$ for images of rectangles (we had to lower the sampling size due to degenerated performance of the metric tree-based $k$-NN search in this particular setup). A single core of AMD EPYC 7543 CPU was used for nearest neighbors search and MI calculation. Each run (one seed) took no longer than one CPU-hour to be completed.

**MINE, NWJ, InfoNCE.** These discriminative approaches are fundamentally alike: each method estimates mutual information by maximizing the associated KL-divergence bound:

$$I(X_0; X_1) = \mathrm{KL}\left(\pi(x_0, x_1) \| \pi(x_0)\pi(x_1)\right) \geqslant \sup_{T \colon \mathbb{R}^d \times \mathbb{R}^d \to \mathbb{R}} \mathrm{F}[T(x_0, x_1)], \tag{25}$$

where $T$ is measurable, and F is some method-specific functional. In practice, $T$ is approximated via a neural network, with the right-hand-side in (25) being used as the loss function.

Motivated by this similarity, we use a nearly identical experimental framework to assess each approach within this category. To approximate $T$ in experiments with synthetic images, we adopt the critic NN architecture from (Butakov et al., 2024a), which we also report in Table 12 ("Critic NN").

The networks were trained via Adam optimizer on $10^5$ images with a learning rate $10^{-3}$, a batch size 512 (with InfoNCE being the only exception, for which we used batch size 256 for training and 512 for evaluation due to memory constraints), and MAE loss for $10^5$ steps. For averaging, we used 5 different seeds. Nvidia A100 was used to train the models. In any setup, each run (one seed) took no longer than two GPU-hours to be completed. The size of discriminator neural network is around 50k parameters.

**AE-WKL Baseline.** Here we include an additional baseline Auto Encoder + Weighted Kozachenko-Leonenko (AE-WKL) for the image based benchmark, which we exclude from the main part of the paper due to instability w.r.t. hyperparameters and huge inductive bias.

The idea of leveraging lossy compression to tackle the curse of dimensionality and provide better MI estimates is well-explored in the literature (Goldfeld & Greenewald, 2021; Goldfeld et al., 2022; Tsur et al., 2023; Fayad & Ibrahim, 2023; Greenewald et al., 2023; Butakov et al., 2024b). In our work, we adopt the non-linear compression setup from (Butakov et al., 2024b), which employs autoencoders for data compression and weighted Kozachenko-Leonenko method (Berrett et al., 2019) for MI estimation in the latent space.

However in this approach non-parametric MI estimator (WKL) approach **heavily relies on the autoencoder latent space**.

First, prior works suggest that non-parametric MI estimators are highly prone to the curse of dimensionality compared to NN-based approaches (Goldfeld et al., 2020; Czyż et al., 2023; Butakov et al., 2024a) and hence WKL performance seriously deteriorates with growth of latent space dimensionality. In particular, Figures 2 and 3 in (Goldfeld et al., 2020) and Table 1 in (Butakov et al., 2024a) indicate that weighted Kozachenko-Leonenko estimator fails to yield reasonable estimates at all if the dimensionality reaches certain threshold.

Second, the performance of estimator depends on the amount of information preserved in the latent space by the autoencoder. If the dimension of latent space is less than dimension of the data manifold, the performance of the method should deteriorate as well.

We validate the AE-WKL approach with different dimensionality of autoencoder latent space in the Gaussian images setup. We increase the latent space dimensionality from $d = 2$ (which is the intrinsic dimensionality of the dataset in question) to $d \in \{4, 5, 8\}$. We report our results in Figure 6. One can see while with $d = 2$ latent space (which is the true latent dimensionality of gaussian images) AE-WKL approach achieves remarkable results, but with growth of dimension to $d \in \{4, 5, 8\}$ the performance starts to deteriorate with $d = 8$ even leading to exploded estimates.

That leaves autoencoder approaches combined with non parametric MI estimators highly dependent on the successful latent space choice. This choice is trivial on the image based benchmark, where the dimensionality of data manifold is known by construction, however such information is usually unknown in real world scenario, which makes approaches like AE-WKL highly unreliable.

The autoencoders were trained using Adam optimizer on $10^5$ images with a batch size $512$, a learning rate $10^{-3}$ and MAE loss for $10^4$ steps. For averaging, we used $5$ different seeds. Nvidia A100 was used to train the autoencoder model. Each run (one seed) took no longer than one GPU-hour to be completed.

**Runtime comparisons**  One can see the approximate runtime comparisons between the featured methods in Table 13.

| Method | Runtime | Hardware |
|---|---|---|
| InfoBridge | 6 hours | GPU |
| MINDE | 2 hours | GPU |
| MIENF | 4 hours | GPU |
| InfoNCE | 1 hour | GPU |
| MINE | 1 hour | GPU |
| NWJ | 1 hour | GPU |
| KSG | 1 hour | CPU |

Table 13: Comparison of methods approximate runtime on the Image benchmark $16 \times 16$. GPU states for NVIDIA A100.

### D.4 PROTEIN EMBEDDINGS DATA

Table 14: The best result is **bolded and underlined** and second best result is **bolded**.

| Method | *Info*Bridge | MINE | InfoNCE | KSG | MINDE-C | MINDE-J |
|---|---|---|---|---|---|---|
| MAE ↓ | **0.04** | 0.223 | 0.237 | **0.116** | 9.29 | 1342 |
| MAE Std | 0.01 | 0.017 | 0.007 | - | 1.1 | 57 |

**Data.**  We use the UniProt database proteins, their embeddings from prottrans_t5_xl_u50 (Elnaggar et al., 2021) model and download them from `https://github.com/ggdna/latent-mutual-information`. All proteins longer than $12 \cdot 10^3$ residues are excluded. These embeddings are then unit variance normalized. The final number of each protein embeddings (samples) used for training is 20641 and the dimensionality is 1024.

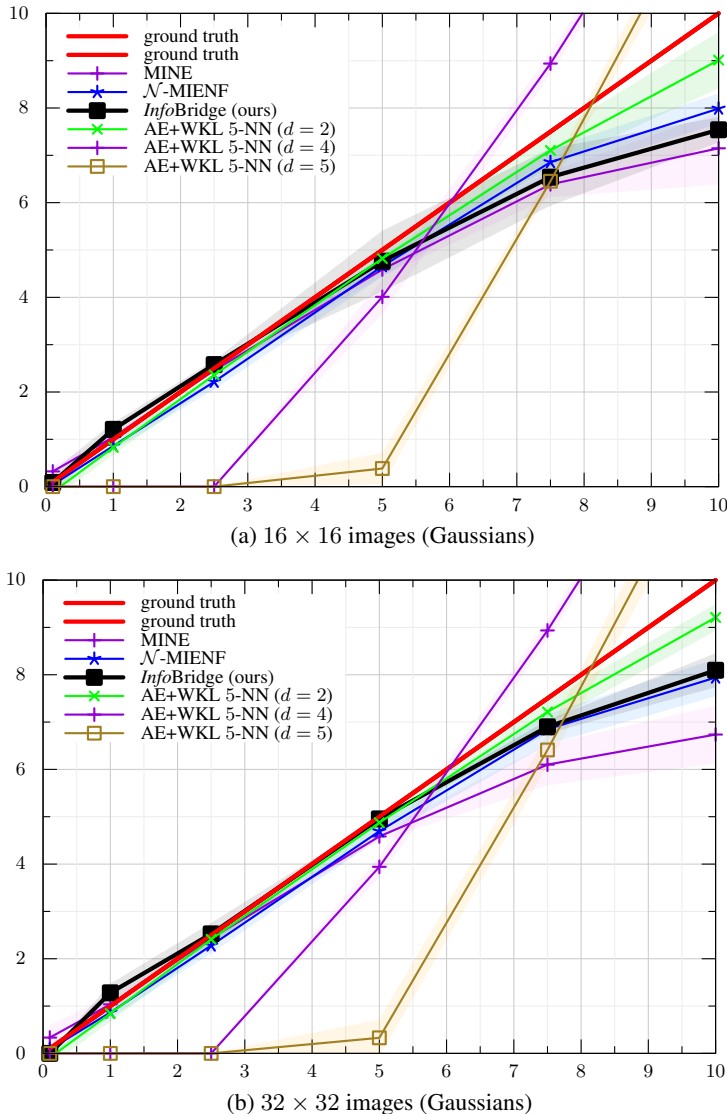

(a) $16 \times 16$ images (Gaussians)

(b) $32 \times 32$ images (Gaussians)

Figure 6: Comparison of the MI estimators. Along $x$ axes is $I(X_0; X_1)$, along $y$ axes is MI estimate $\hat{I}(X_0; X_1)$. We plot 99% confidence intervals acquired from different seed runs.

**Methods implementation.**

*Info***Bridge**. Hyperparameters are the same as for low-dimensional benchmark Appendix D.2, MLP has $\sim 900k$ parameters, weight decay is $0.001$, dropout is $0.2$ and the model is trained for 100 epochs. Results for InfoBridge averaged over last 5 MI estimations during training. The size of neural network is around 900k parameters. The time $t$ is sampled uniformly in $t \in [0, 1 - 10^{-3})$ segment.

**MINDE**. We leverage the official implementation of MINDE `https://github.com/MustaphaBounoua/minde` in the configuration used for the low-dimensional benchmark, with $\sim 900k$ parameters MLP, lr $1e - 4$, batch size of $256$, ema decay of $0.999$ and train model for 100 epochs.

**MINE, InfoNCE**. These estimators are taken from the *bmi* library `https://github.com/cbg-ethz/bmi` with all the default parameters. The size of discriminator neural network is around 1k parameters.

**KSG**. Implementation is taken from `https://github.com/ggdna/latent-mutual-information` and used with all the default parameters on 20000 training samples.

All the neural methods were trained on Nvidia A100 and took less then one GPU-hour to complete. KSG estimation algorithm took approximately one hour to complete.

**Runtime comparisons.**    One can see the approximate runtime comparisons between the featured methods in Table 15.

| Method | Runtime | Hardware |
|---|---|---|
| InfoBridge | 20 minutes | GPU |
| MINDE | 3 minutes | GPU |
| MINE | 1 minute | CPU |
| InfoNCE | 1 minute | CPU |
| KSG | 20 minutes | CPU |

Table 15: Comparison of methods approximate runtime on the Protein Data experiments. GPU states for NVIDIA A100.

## D.5    HIGH MUTUAL INFORMATION EXPERIMENTS

**Half Cube transform.**    We adopt the half-cube transform for Gaussian random variables as introduced in (Czyż et al., 2023):

$$f(x) = x\sqrt{|x|}.$$

This transform lengthens the tails of Gaussian random variables, while preserving the MI due to $f(x)$ transform being reversible.

**Additional experimental results.**    In addition to the high MI experiment in the main text (§ 5.4) here we present extended comparisons with other baselines: see Table 16 for Gaussian random variables, Table 17 for Gaussian random variables with half cube transform, Table 18 for Correlated Uniform random variables, Table 19 for Smoothed Uniform random variables. Overall our method still is the most reliable and accurate of the bunch.

As shown, our method performs well with both $\epsilon = 1$ and $\epsilon = 0.01$, with the latter providing slightly better results. For this reason, we adopt $\epsilon = 0.01$ in the main text comparisons.

One can see that MINDE-J offers the closest to the ground truth MI in many setups, but we note that it ***drastically overestimates*** the MI in the d=160, MI=80 setup across all the distribution types. Such behaviour is typical for MINDE-J on complex datasets also, see (§ 5.2) and (§ 5.3) that is why we leave it out in the main text comparisons in (§ 5.4).

| Method | d=20, MI=10 | d=40, MI=20 | d=80, MI=40 | d=160, MI=80 |
|---|---|---|---|---|
| InfoBridge (**ours**, $\epsilon = 1$) | **9.85** | **19.99** | **39.31** | 78.86 |
| InfoBridge (**ours**, $\epsilon = 0.01$)[†] | 10.35 | 21.79 | 41.92 | **79.76** |
| MINDE-C[†] | 9.59 | 16.96 | 35.51 | 56.27 |
| MINDE-J | 10.23 | 20.02 | 42.11 | 450 |
| MINE[†] | 6.51 | 8.43 | 7.76 | 8.08 |
| InfoNCE[†] | 5.14 | 5.40 | 4.76 | 3.80 |
| fDIME (KL, J) | 5.51 | | 2.49 | 0.0 |
| fDIME (HD, J) | 5.24 | 3.66 | 1.14 | 0.0 |
| fDIME (GAN, J)[†] | 6.88 | 6.71 | 3.54 | 0.0 |
| fDIME (KL, D) | 3.71 | 4.06 | 0.22 | 0.0 |
| fDIME (HD, D) | 4.01 | 4.24 | 0.23 | 0.0 |
| fDIME (GAN, D) | 4.5 | 4.86 | 0.32 | 0.0 |

Table 16: Mutual Information estimates across increasing dimensions and mutual information levels for the Gaussian random variables. Best MI estimation is **bolded** and second best is underlined. [†] stands for method shown in the high MI experiment in the main text. Empty cell stands for method explosion.

**Experimental Details.**    We use 100k train samples dataset and 10k test dataset for all the methods. Datasets are generated using *mutinfo* library github repository:

| Method | d=20, MI=10 | d=40, MI=20 | d=80, MI=40 | d=160, MI=80 |
|---|---|---|---|---|
| InfoBridge (**ours**, $\epsilon = 1$) | 9.07 | 18.23 | 34.35 | 61.63 |
| InfoBridge (**ours**, $\epsilon = 0.01$)[†] | **9.92** | **20.69** | **39.6** | **84.24** |
| MINDE-C[†] | 8.31 | 14.69 | 33.64 | 40.63 |
| MINDE-J | 9.28 | 16.49 | 43.3 | 308 |
| MINE[†] | 3.18 | 3.0 | 3.28 | 3.74 |
| InfoNCE[†] | 4.11 | 4.7 | 3.84 | 2.74 |
| fDIME (KL, J) | 7.52 | 6.84 | 11.3 | 8.18 |
| fDIME (HD, J) | 6.56 | 11.59 | 10.77 | 10.73 |
| fDIME (GAN, J)[†] | 9.7 | 14.7 | 20.84 | 13.91 |
| fDIME (KL, D) | 5.41 | 6.82 | 6.67 | 6.76 |
| fDIME (HD, D) | 5.71 | 7.66 | 7.99 | 8.74 |
| fDIME (GAN, D) | 7.83 | 10.4 | 11.49 | 10.86 |

Table 17: Mutual Information estimates across increasing dimensions and mutual information levels for the Half Cube transformed Gaussian random variables. Best MI estimation is **bolded** and second best is underlined. [†] stands for method shown in the high MI experiment in the main text.

| Method | d=20, MI=10 | d=40, MI=20 | d=80, MI=40 | d=160, MI=80 |
|---|---|---|---|---|
| InfoBridge (**ours**, $\epsilon = 1$) | 7.95 | 15.15 | 28.46 | 41.9 |
| InfoBridge[†] (**ours**, $\epsilon = 0.01$) | 8.65 | 16.93 | 33.69 | **57.73** |
| MINDE-C[†] | 8.03 | 15.14 | 30.78 | 55.91 |
| MINDE-J | 8.8 | **17.56** | **34.59** | 539 |
| MINE[†] | 6.12 | 7.9 | 7.92 | 7.11 |
| InfoNCE[†] | 4.8 | 4.48 | 3.71 | 3.42 |
| fDIME (KL, J) | 7.6 | | 6.1 | 0.99 |
| fDIME (HD, J) | 8.28 | 8.84 | 4.8 | 0.19 |
| fDIME (GAN, J)[†] | **9.2** | 10.33 | 8.75 | 1.56 |
| fDIME (KL, D) | 5.62 | 5.1 | 2.71 | 0.1 |
| fDIME (HD, D) | 6.6 | 7.02 | 2.14 | 0.41 |
| fDIME (GAN, D) | 6.49 | 7.88 | 3.79 | 0.0 |

Table 18: Mutual Information estimates across increasing dimensions and mutual information levels for the Correlated Uniform random variables. Best MI estimation is **bolded** and second best is underlined. [†] stands for method shown in the high MI experiment in the main text.

https://github.com/VanessB/mutinfo.

***Info*Bridge.** Implementation was taken with exactly the same hyperparameters as for low-dimensional benchmark experiment, see Appendix D.2.

**MINDE.** We leverage the official implementation of MINDE https://github.com/MustaphaBounoua/minde in the configuration used for the low-dimensional benchmark, with $\sim 900k$ parameters MLP, lr $1e - 4$, batch size of 256, ema decay of 0.999 and train model for 100 epochs.

**MINE, InfoNCE**. These estimators are taken from the *bmi* library https://github.com/cbg-ethz/bmi with all the default parameters. The size of discriminator neural network is around 1k parameters.

**fDIME.** implementation was taken from the official github repository:

https://github.com/tonellolab/fDIME,

with all the default hyperparameters. *KL, HD, GAN* stand for different divergences and *J, D* stand for Joint and Derranged architectures of fDIME.

**Runtime comparisons.** One can see the approximate runtime comparisons between the featured methods in Table 20.

| Method | d=20, MI=10 | d=40, MI=20 | d=80, MI=40 | d=160, MI=80 |
|---|---|---|---|---|
| InfoBridge (**ours**, $\epsilon = 1$) | 6.4 | 12.83 | 24.21 | 29.83 |
| InfoBridge (**ours**, $\epsilon = 0.01$)[†] | 7.3 | **14.63** | 29.25 | **50.28** |
| MINDE-C[†] | 6.53 | 13.23 | 24.98 | 44.2 |
| MINDE-J | 7.63 | 14.39 | **31.15** | 511 |
| MINE[†] | 5.56 | 7.45 | 7.91 | 7.17 |
| InfoNCE[†] | 3.84 | 3.67 | 3.34 | 3.14 |
| fDIME (KL, J) | 6.15 | -74 | 3.09 | 0.0 |
| fDIME (HD, J) | 5.9 | 8.12 | 3.95 | 0.0 |
| fDIME (GAN, J)[†] | **7.75** | 8.49 | 4.75 | 0.0 |
| fDIME (KL, D) | 4.91 | 4.17 | 0.32 | 0.0 |
| fDIME (HD, D) | 4.59 | 4.82 | 0.37 | 0.0 |
| fDIME (GAN, D) | 5.65 | 5.65 | 0.33 | 0.0 |

Table 19: Mutual Information estimates across increasing dimensions and mutual information levels for the Smoothed Uniform random variables. Best MI estimation is **bolded** and second best is underlined. [†] stands for method shown in the high MI experiment in the main text.

| Method | Runtime | Hardware |
|---|---|---|
| InfoBridge | 30 minutes | GPU |
| MINDE | 10 minutes | GPU |
| f-DIME | 1 minute | GPU |
| MINE | 1 minute | CPU |
| InfoNCE | 1 minute | CPU |

Table 20: Comparison of methods approximate runtime on the High MI experiments in the $D = 160$, MI=80 case. GPU states for NVIDIA A100.

Table 21 (rotated landscape). Methods (rows) with their Method Type grouping, evaluated across 40 datasets (columns). Values are mean estimates; the two leftmost label columns give the method and its type.

| Method | Type | Wiggly @ Bivariate Nm 1×1 | Uniform 1×1 (noise=.75) | Uniform 1×1 (noise=.1) | Swiss roll 2×1 | St 5×5 (dof=3) | St 5×5 (dof=2) | St 3×3 (dof=3) | St 3×3 (dof=2) | St 2×2 (dof=2) | St 2×2 (dof=1) | St 1×1 (dof=1) | Sp@Nm CDF@Mn 5×5 | Sp@Nm CDF@Mn 3×3 | Sp@Nm CDF@Mn 25×25 | Sp@Mn 5×5 | Sp@Mn 3×3 | Sp@Mn 25×25 | Nm CDF@Mn 5×5 | Nm CDF@Mn 3×3 | Nm CDF@Mn 25×25 | Nm CDF@Bivariate Nm 1×1 | Mn 50×50 (dense) | Mn 5×5 (dense) | Mn 5×5 (2-pair) | Mn 3×3 (dense) | Mn 3×3 (2-pair) | Mn 25×25 (dense) | Mn 25×25 (2-pair) | Mn 2×2 (dense) | Mn 2×2 (2-pair) | Hc@Mn 5×5 | Hc@Mn 3×3 | Hc@Mn 25×25 | Hc@Bivariate Nm 1×1 | Bivariate Nm 1×1 | Bimodal 1×1 | Asinh@St 5×5 (dof=2) | Asinh@St 3×3 (dof=2) | Asinh@St 2×2 (dof=1) | Asinh@St 1×1 (dof=1) |
|---|---|---|---|---|---|---|---|---|---|---|---|---|---|---|---|---|---|---|---|---|---|---|---|---|---|---|---|---|---|---|---|---|---|---|---|---|---|---|---|---|---|
| GT | Method Type | 0.22 | 0.43 | 0.29 | 0.45 | 0.41 | 0.41 | 0.40 | 0.40 | 0.38 | 0.95 | 0.96 | 0.98 | 0.29 | 0.41 | 1.02 | 0.29 | 0.41 | 1.02 | 1.02 | 1.02 | 0.41 | 1.02 | 1.02 | 0.59 | 1.02 | 0.59 | 1.02 | 1.02 | 0.41 | 1.02 | 0.99 | 0.41 | 1.00 | 0.97 | 0.96 | 0.92 | 0.91 | 0.96 | 0.97 | 0.96 |
| *Info*Bridge | Bridge Matching | 0.26 | 0.48 | 0.31 | 0.45 | 0.44 | 0.40 | 0.40 | 0.38 | 0.95 | 0.96 | 0.98 | 0.29 | 0.41 | 1.02 | 0.29 | 0.41 | 1.02 | 1.02 | 1.02 | 0.41 | 1.02 | 1.02 | 0.59 | 1.02 | 0.59 | 1.02 | 1.02 | 0.41 | 1.02 | 0.99 | 0.41 | 1.00 | 0.97 | 0.96 | 0.92 | 0.91 | 0.96 | 0.97 | 0.96 | |
| NVF | Flow | 0.18 | 0.40 | 0.30 | 0.55 | 0.41 | 0.41 | 0.96 | 1.01 | 1.00 | 0.95 | 0.97 | 1.02 | 0.80 | 0.48 | 0.58 | 0.90 | 0.97 | 1.01 | 1.52 | 0.40 | 0.95 | 1.72 | 0.40 | 0.96 | 1.73 | 0.41 | 1.07 | 1.62 | 0.40 | 0.94 | 1.60 | 0.40 | 0.98 | 1.28 | 0.87 | 0.66 | 1.03 | 0.29 | 1.27 | 0.65 |
| JVF | Flow | 0.0 | 0.0 | 0.0 | 0.0 | 0.41 | 0.41 | 1.02 | 1.02 | 1.02 | 1.02 | 1.02 | 0.59 | 2.66 | 0.01 | 0.41 | 1.02 | 1.02 | 0.87 | 0.90 | 0.95 | 0.41 | 1.02 | 1.02 | 0.83 | 0.27 | 0.41 | 1.65 | 0.41 | 1.02 | 0.83 | 0.27 | 0.41 | 0.95 | 0.64 | 0.94 | 1.27 | 16.11 | 0.37 | 0.95 | 0.63 |
| MINDE-j (σ = 1) | Diffusion | 0.21 | 0.40 | 0.26 | 0.40 | 0.41 | 0.41 | 1.10 | 1.01 | 1.00 | 0.42 | 1.00 | 0.59 | 1.72 | 0.40 | 0.96 | 0.99 | 0.90 | 0.95 | 0.95 | 0.98 | 0.17 | 0.35 | 0.18 | 0.25 | 0.17 | 0.47 | 0.29 | 0.49 | 1.65 | 0.31 | 0.41 | | | | | | | | | |
| MINDE-j | Diffusion | 0.22 | 0.42 | 0.28 | 0.42 | 0.41 | 0.42 | 1.19 | 1.02 | 1.02 | 0.99 | 1.31 | 1.02 | 1.73 | 0.41 | 1.07 | 0.99 | 0.99 | 0.95 | 0.92 | 0.93 | 0.13 | 0.24 | 0.20 | 0.30 | 0.18 | 0.48 | 0.31 | 0.40 | 1.67 | 0.31 | 0.42 | | | | | | | | | |
| MINDE-c (σ = 1) | Diffusion | 0.21 | 0.42 | 0.27 | 0.42 | 0.41 | 0.41 | 0.96 | 1.00 | 0.99 | 1.00 | 1.26 | 1.01 | 1.62 | 0.40 | 0.94 | 0.93 | 0.96 | 0.94 | 0.13 | 0.28 | 0.18 | 0.29 | 0.18 | 0.42 | 0.30 | 0.32 | 1.67 | 0.31 | 0.41 | | | | | | | | | | | |
| MINDE-c | Diffusion | 0.21 | 0.42 | 0.28 | 0.42 | 0.41 | 0.41 | 1.00 | 1.01 | 1.01 | 1.01 | 1.27 | 1.01 | 1.60 | 0.40 | 0.98 | 0.99 | 0.98 | 0.92 | 0.94 | 0.98 | 0.14 | 0.26 | 0.19 | 0.28 | 0.17 | 0.44 | 0.29 | 0.40 | 1.66 | 0.31 | 0.41 | | | | | | | | | |
| MINE | Classical | 0.23 | 0.38 | 0.24 | 0.36 | 0.40 | 0.41 | 0.41 | 0.96 | 0.99 | 0.98 | 1.01 | 0.30 | 0.99 | 1.28 | 1.01 | 1.00 | 0.59 | 1.60 | 0.39 | 0.88 | 0.90 | 0.81 | 0.70 | 0.65 | 0.88 | 0.89 | 0.87 | 0.02 | 0.01 | 0.12 | 0.12 | 0.13 | 0.16 | 0.22 | 0.39 | 1.66 | 0.32 | 0.41 | | |
| InfoNCE | | 0.22 | 0.41 | 0.27 | 0.40 | 0.41 | 0.41 | 0.41 | 0.98 | 1.01 | 1.01 | 1.02 | 0.59 | 1.61 | 0.40 | 0.92 | 0.98 | 0.99 | 0.83 | 0.84 | 0.82 | 0.96 | 0.96 | 0.15 | 0.30 | 0.18 | 0.27 | 0.17 | 0.41 | 0.28 | 0.40 | 1.69 | 0.32 | 0.41 | | | | | | | |
| D-V | | 0.22 | 0.41 | 0.27 | 0.40 | 0.41 | 0.41 | 0.41 | 0.98 | 1.01 | 1.01 | 1.02 | 0.59 | 1.61 | 0.40 | 0.93 | 0.98 | 0.99 | 0.82 | 0.82 | 0.81 | 0.92 | 0.96 | 0.01 | 0.05 | 0.11 | 0.13 | 0.15 | 0.22 | 0.21 | 0.40 | 1.69 | 0.32 | 0.41 | | | | | | | |
| NWJ | | 0.22 | 0.40 | 0.27 | 0.40 | 0.41 | 0.41 | 0.41 | 0.98 | 1.01 | 1.01 | 1.02 | 0.59 | 1.60 | 0.40 | 0.93 | 0.98 | 0.98 | 0.82 | 0.82 | 0.80 | 0.92 | 0.95 | 0.03 | 0.02 | 0.04 | -0.65 | 0.12 | 0.21 | 0.40 | 1.69 | 0.32 | 0.41 | | | | | | | | |
| KSG | Classical | 0.22 | 0.38 | 0.19 | 0.24 | 0.42 | 0.42 | 0.17 | 0.66 | 1.03 | 0.29 | 0.91 | 0.95 | 0.41 | 0.74 | 0.57 | 1.28 | 0.20 | 0.18 | 0.71 | 0.55 | 0.20 | 0.90 | 0.69 | 0.16 | 0.22 | 0.09 | 0.12 | 0.07 | 0.20 | 0.15 | 0.42 | 1.68 | 0.32 | 0.42 | | | | | | |
| LNN | | 0.25 | 0.89 | 2.71 | 6.65 | 0.41 | 0.42 | 0.42 | 2.68 | 6.45 | 0.39 | 2.49 | 7.31 | 6.76 | 0.39 | 3.10 | 2.49 | 2.38 | 7.24 | 0.53 | 1.20 | 2.12 | 2.74 | 5.48 | 0.34 | 0.48 | 0.29 | 0.42 | | | | | | | | | | | | | |
| CCA | | 0.00 | 0.00 | 0.00 | 0.00 | 0.34 | 0.41 | 0.38 | 0.99 | 0.95 | 0.96 | 1.02 | 0.29 | 1.06 | 1.33 | 1.02 | 0.41 | 1.00 | 0.97 | 0.96 | 0.86 | 0.23 | 0.39 | 1.75 | 0.39 | 1.00 | 0.97 | 0.96 | 0.95 | 0.11 | 0.13 | 0.01 | 0.31 | 0.02 | 1.63 | 0.19 | 0.38 | | | | |
| DoE(Gaussian) | | 0.16 | 0.48 | 0.27 | 0.57 | 0.37 | 0.39 | 0.67 | 7.83 | 0.95 | 0.64 | 0.94 | 1.27 | 16.11 | 0.37 | 0.99 | 0.95 | 0.84 | 0.90 | 0.99 | 0.82 | 0.81 | 0.80 | 0.48 | 0.58 | 0.56 | 0.57 | 0.74 | 0.77 | 6.74 | 7.93 | 1.78 | 2.54 | 0.61 | 4.24 | 1.17 | 1.57 | 0.11 | 0.38 | | |
| DoE(Logistic) | | 0.13 | 0.37 | 0.21 | 0.43 | 0.41 | 0.35 | 0.36 | 0.62 | 7.83 | 0.92 | 0.95 | 0.92 | 0.95 | 0.63 | 0.93 | 1.27 | 16.15 | 0.41 | 0.78 | 1.05 | 0.47 | 0.60 | 0.55 | 0.67 | 0.79 | 0.81 | -0.32 | 2.00 | 0.46 | 0.82 | 0.29 | 1.48 | 0.59 | 1.58 | 0.08 | 0.35 | | | | |
| dist | | | | | | | | | | | | | | | | | | | | | | | | | | | | | | | | | | | | | | | | | |

Table 21: Mean estimate over 10 seeds using $10 \cdot 10^4$ test samples compared each against the ground-truth. Size of train dataset for every neural method is 100k. All the methods for comparison, except for NVF and JVF Dahlke & Pacheco (2025), with *Info*Bridge were taken from Franzese et al. (2024). List of abbreviations (*Mn*: Multinormal, *St*: Student-t, *Nm*: Normal, *Hc*: Half-cube, *Sp*: Spiral)

