# OpenReview forum: "InfoBridge: Mutual Information estimation via Bridge Matching"
_ICLR.cc/2026/Conference — ICLR 2026 Poster_

### Official Review · Reviewer_UxJn · 2025-10-22

**Soundness:** 3
**Presentation:** 3
**Contribution:** 3
**Rating:** 6
**Confidence:** 4

**Summary:**

The paper introduces a new formulation of mutual information (MI) estimation as a domain transfer problem. In particular, it estimates MI by quantifying the difference between the drift functions of diffusion bridge processes associated with the joint distribution and the product of marginal (independent) distributions of the variables. The idea is interesting and improves previously studied diffusion-based estimators, to which the authors compare by running extensive benchmarks.

**Strengths:**

- The authors theoretically validate the correctness of their new approach to use Bridge Matching for MI estimation.
- The difference with prior work on diffusion-based estimator MINDE is well-expressed and theoretically grounded.
- A rich set of standard benchmarks and ablation studies has been conducted.

**Weaknesses:**

- The main weakness in the experimental evaluation is that the authors only consider very large sample sizes; 100k training datapoints, while classical approaches have also been tested for 5k or less. Although the main competitor, MINDE, also used 100k training points, it would be interesting to see if the method can also perform well with significantly fewer samples, e.g., 10k.
- While the MINDE estimator requires only the learned score functions to estimate mutual information, InfoBridge relies on full sampling of two diffusion bridge trajectories to approximate the drifts. It is not clear if this has implications for computational cost and scalability. It would be relevant for the authors to address and discuss the computational cost difference between the two methods.
- The paper refers several times to regularity conditions, such as the existence of finite first moments, as prerequisites for the main theorem (Theorem 4.1). However, these assumptions are not stated explicitly in the main text nor appendix. Since these are key hypotheses for the validity of the theoretical results, it would be important to summarize them clearly.
- The volatility coefficient $\epsilon$ is studied in the 16×16 Gaussian and protein embedding experiments, leading to a rule-of-thumb setting $\epsilon = 1$. However, the paper does not provide a detailed analysis of how $\epsilon$ influences the estimator’s bias and variance, nor a principled procedure for selecting it.
- Methods for MI estimation between random variables of different dimension are mentioned but not considered for experiments.
- As mentioned by the authors, heavy-tailed distributions cannot be modelled well.

Minor remarks:

- Some formulas and tables violate the space constraints (see e.g., page 3).
- Some references in the appendix are not rendered correctly.
- Reference: lee & Rhee missing year.

**Questions:**

- For the experiment in Table 1, some methods do not require training. Do you only provide them with 10k test samples, or do they get access to 110k samples, i.e., the full train and test split. How would the method perform with only 10k samples for training and testing jointly?
- Do you have an intuition as of why higher MI values seem to lead to worst estimation in the image gaussian 16x16 case, for all estimators?
- “First, by leveraging finite-time bridge processes instead of infinite-time diffusion (which in practice is approximated with a finite-time horizon), we provide an unbiased estimator.” Why does discretization on the bridges process not induce a bias while it does for noise-to-data diffusion?

---

> ### Author Response · Authors · 2025-11-22
> **Rebuttal**
>
> Dear reviewer **UxJn**, we thank you for taking the time to review our paper. Let us here answer your questions.
>
> **[W1] The main weakness in the experimental evaluation is that the authors only consider very large sample sizes; 100k training datapoints, while classical approaches have also been tested for 5k or less. Although the main competitor, MINDE, also used 100k training points, it would be interesting to see if the method can also perform well with significantly fewer samples, e.g., 10k.**
>
> **Answer**: To answer your question we have performed study on the number of training samples. We present it in **Appendix C.3** of the revised version. In particular, we vary the **train sample size** in the Image Data experiment Section 5.2 with Rectangle images of 16x16 size. We test MINE, MINDE, and InfoBridge on 5k, 10k, and 100k train sample sizes with a 10k test sample size. To prevent overfitting, we lower the number of parameters in the neural network for MINDE and InfoBridge. The neural network sizes are provided in the Table. All the other hyperparameters are exactly as previously. Methods on 5k and 10k train sample sizes are run with 1 seed, and 100k train sample size results were taken from the main paper. MAE means Mean Absolute Error. The best MAE for each training sample size is **bolded**.
>
> | Method | N samples | N params | MAE | 0.1 | 1 | 2.5 | 5 | 7.5 | 10 |
> | --- | --- | --- | --- | --- | --- | --- | --- | --- | --- |
> | InfoBridge (ours) | 5k | 106k | **0.199** | 0.008 | 0.79 | 2.33 | 4.83 | 7.2 | 9.75 |
> | MINDE | 5k | 106k | 2.397 | 0.47 | 0.36 | 0.44 | 0.38 | 4.9 | 5.91 |
> | MINE | 5k | 50k | 1.618 | 0.0 | 0.6 | 2.0 | 3.74 | 4.77 | 5.28 |
> | InfoBridge (ours) | 10k | 428k | **0.142** | 0.006 | 1.02 | 2.65 | 5.24 | 7.65 | 9.8 |
> | MINDE | 10k | 428k | 1.663 | 0.46 | 0.44 | 0.48 | 5.93 | 6.93 | 4.46 |
> | MINE | 10k | 50k | 1.513 | 0.0 | 0.78 | 2.1 | 3.9 | 4.85 | 5.39 |
> | InfoBridge (ours) | 100k | 27M | **0.228** | 0.0 | 1.16 | 2.79 | 5.47 | 7.67 | 9.82 |
> | MINDE | 100k | 27M | 0.815 | 0.41 | 0.91 | 2.45 | 4.22 | 5.63 | 8.21 |
> | MINE | 100k | 50k | 0.978 | 0.47 | 1.15 | 2.43 | 4.41 | 5.86 | 6.95 |
>
> One can see that our method retains its performance on a lower sample size, while MINDE and MINE start to degrade.
>
> **[W2]  While the MINDE estimator requires only the learned score functions to estimate mutual information, InfoBridge relies on full sampling of two diffusion bridge trajectories to approximate the drifts. It is not clear if this has implications for computational cost and scalability. It would be relevant for the authors to address and discuss the computational cost difference between the two methods.**
>
> **Answer**: We would like to note that **InfoBridge does not rely on full sampling of diffusion bridge trajectories,** nor during training or inference. Instead, we require only one time $t$ since of diffusion bridge trajectories per gradient step. One can see that the optimization problems stated in the Conditional Bridge Matching paragraph Section 2 do not depend on the trajectory but rather on independent time $t$ slices of the trajectory.
>
> **InfoBridge training and inference procedures are simulation-free.** In that sense, the computational complexity of training diffusion models is the same as for training the bridge matching models. Overall **computational complexity of training and inference** of MINDE and InfoBridge should be **comparable**.
>
> We include the **wall clock comparisons in Appendix D.** Our method is a bit slower than MINDE, but we attribute it to the differences in implementation. InfoBridge is implemented in native PyTorch, while MINDE is implemented in PyTorch Lightning.
>
> To foster readability, we have added more explicit statements on the sufficiency of time slices $t$ of Brownian Bridge processes rather than full trajectories, see Sections 2 and 4 in the revised version.
>
> **[W3] The paper refers several times to regularity conditions, such as the existence of finite first moments, as prerequisites for the main theorem (Theorem 4.1). However, these assumptions are not stated explicitly in the main text nor appendix. Since these are key hypotheses for the validity of the theoretical results, it would be important to summarize them clearly.**
>
> **Answer**: We have added an explicit list of assumptions in **Appendix A** of the revised version.
>
> **[W4] Methods for MI estimation between random variables of different dimension are mentioned but not considered for experiments.**
>
> Estimation of MI between variables of different dimensions can be reduced to the estimation of MI between variables of the same dimension by methods described in **Appendix B.5** of the revised version (Appendix C.5 of the original submission). At the moment we leave this as a future work and concentrate on MI estimation between random variables of the same dimension, as this methodology can be directly translated onto the task of estimation of MI between variables of different dimensionality.

---

> ### Author Response · Authors · 2025-11-22
> **Continuation of the rebuttal**
>
> **[W5] The volatility coefficient $\epsilon$ is studied in the 16×16 Gaussian and protein embedding experiments, leading to a rule-of-thumb setting $\epsilon = 1$. However, the paper does not provide a detailed analysis of how $\epsilon$ influences the estimator’s bias and variance, nor a principled procedure for selecting it.**
>
> **Answer**: In general, we would say that $\epsilon$ hyperparameter is a tool for **regularization**. The **bigger** $\epsilon$ hyperparameter corresponds to the **reduction of variance and growth of stability**, i.e., the learning procedure is very stable but slow and the method is stable w.r.t. other hyperparameters. While a **smaller** $\epsilon$ hyperparameter corresponds to the **growth of variance and reduction in stability**, i.e., the learning procedure has more variance but converges faster, and the model is less stable w.r.t. other hyperparameters. The general advice is to keep $\epsilon$ as **low as possible to keep the learning procedure stable**. In most cases, **$\epsilon=1$ is enough** to get decent results and keep the model stable. We have added the corresponding discussion to **Appendix C**.
>
> **[W6] As mentioned by the authors, heavy-tailed distributions cannot be modelled well.**
>
> **Answer**: We would like to add that we have tested our model on many heavy-tailed distributions, including student distributions in **Section 4.1** and half-cubed transformed Gaussian in **Section 4.2**. Our method struggles only on Student distributions with a single degree of freedom, which has no first moment, while on **other heavy-tailed distributions**, our method performs **quite well**.
>
> We would like to repeat that  Student distributions with a single degree of freedom present a theoretical challenge to our method due **theoretical requirements of  Bridge Matching**, but we are **able to overcome** this problem by applying the **tail shortening transform *asinh**.* The results are shown in the first two columns of Table 1.
>
> **[W7] Some formulas and tables violate the space constraints (see e.g., page 3).**
>
> **Answer**: We thank the reviewer for noting it. We have fixed the space constraints violations in the main text and the appendix.
>
> **[W8] Some references in the appendix are not rendered correctly.**
>
> **Answer:** We have fixed the not properly rendered references in the Appendix.
>
> **[W9] Reference: lee & Rhee missing year.**
>
> **Answer:** We have added a year to this citation.
>
> **[Q1] For the experiment in Table 1, some methods do not require training. Do you only provide them with 10k test samples, or do they get access to 110k samples, i.e., the full train and test split. How would the method perform with only 10k samples for training and testing jointly?**
>
> **Answer:** Our data setup for the methods that do not require training follows [Czyz’24], where these methods do have access to the full combination of train and test split, i.e., 110k samples. We assume that in the case where these methods have only 10k samples, their performance would drop. We would come back with a particular measurement later in the rebuttal.
>
> **[Q2] Do you have an intuition as of why higher MI values seem to lead to worst estimation in the image gaussian 16x16 case, for all estimators?**
>
> **Answer:** The Gaussian blobs dataset has an intrinsic dimensionality of 2, while the rectangles dataset has a dimensionality of 4. Consequently, for the same ground-truth MI, the MI per latent dimension is higher for the Gaussians. This leads to minor numerical instabilities, especially at high MI values where the relationship is almost deterministic and thus challenging for estimators to capture.
>
> **[Q3] “First, by leveraging finite-time bridge processes instead of infinite-time diffusion (which in practice is approximated with a finite-time horizon), we provide an unbiased estimator.” Why does discretization on the bridges process not induce a bias, while it does for noise-to-data diffusion?**
>
> **Answer:**  **Finite-time** **InfoBridge** processes allow us to get access to the full trajectory, while the **infinite-time** process used is **MINDE**. In practice,  the final time of the MINDE process in $T < \infty$, which adds a **bias** into their formulation.
>
> In particular, the MINDE bias term reduces with $T \rightarrow \infty$, but is never zero, while InfoBridge, due to its finite-time formulation, doesn’t have this problem.
>
> **Concluding remarks.** We would be grateful if you could let us know if the explanations we gave have been satisfactory in addressing your concerns and questions about our work. If so, we kindly ask that you consider increasing your rating. We are also open to discussing any other questions you may have.
>
> **Citations:**
>
> [Czyz’24] Paweł Czyz, Frederic Grabowski, Julia E Vogt, Niko Beerenwinkel, and Alexander Marx. Beyond
> normal: On the evaluation of mutual information estimators. In Thirty-seventh Conference on
> Neural Information Processing Systems, 2023

---

> > ### Comment · Reviewer_UxJn · 2025-11-27
> >
> > Thank you for the detailed response and the additional results. The rebuttal confirms my positive impression of the paper. I updated my score accordingly.

---

### Official Review · Reviewer_uyVy · 2025-10-24

**Soundness:** 2
**Presentation:** 1
**Contribution:** 2
**Rating:** 4
**Confidence:** 4

**Summary:**

The paper presents a novel unbiased MI estimator based on diffusion bridge models. The proposed estimator is tested on different low-dimensional and high-dimensional datasets.

**Strengths:**

- The proposed estimator relies on diffusion bridge models, thus leveraging a new technique for MI estimation
- The proposed generative MI estimator is unbiased, differently from MINDE which is biased
- The experimental results show the effectiveness of the proposed estimator

**Weaknesses:**

- Although the proposed estimator leverages diffusion bridge models, which differ from diffusion models, the proposed framework appears as an extension of MINDE
- The paper is not well-written. The clarity could definitively be improved. For instance, SDE is not defined (stochastic differential equation). The equation in line 145 should be written in two lines, as it exceeds the borders of the paper. There are many typos: line 96 (conditioned its on values), line 264 (as a domain transfer task), multiple wrong references to equation (2) while I assume the authors wanted to refer to other equations. The same wrong reference to (2) happens also in line 300. Table 2 reports Mean Average Error, but I assume the authors used the Mean Absolute Error.
- In the experiments  the authors compare the proposed method with f-DIME. However, there is no reference of f-DIME in the related work section, so it should be inserted in the Discriminative estimators paragraph.

**Questions:**

- What is $\delta()$ in equation (3)? It is not defined.
- In line 148 the authors refer to Problem (2). Does it refer to the second equation? That would be the MI definition. In the line below the authors write “minimizes (2)”, but they are not referring to the minimization of MI, I guess they are referring to the minimization of the long equation in line 145.
- In the pseudocode of algorithm 1 and in the code provided in the supplementary material, the authors perform a random permutation to obtain the samples from the product of marginal densities:
y_samples_permuted = y_samples[torch.randperm(y_samples.shape[0])].
However, using such a function leads a certain number of elements of the shuffled vector to be positioned in the same initial position (before the permutation). This strategy was highlighted in f-DIME’s paper as problematic and in that specific case the authors showed that using a specific  discriminative architecture that mistake would lead to an upper bound in the MI estimate. For InfoBridge I do not see any upper bound. However, performing such a random permutation is logically incorrect and the authors should perform the experiments after correcting such a mistake. How does the performance of the proposed algorithm change after this correction? If it does not change, why?
- Where are the experiments justifying the sentence in lines 314-315 about the additional binary input?
- Can the authors show a comparison of the computational complexity between the different algorithms used?
- Why the reference Lee and Rhee has no year?
- The discriminative models used during comparisons have the same size (number of parameters) of the generative estimators?
- Isn’t 100k train samples a very large number? How do the algorithms perform with a lower number of training samples?
- In Figure 2 the authors specify which version of MINDE they are using, but do not specify which f-DIME version. Can the authors specify it?

---

> ### Author Response · Authors · 2025-11-22
> **Rebuttal**
>
> Dear reviewer **uyVy**, we thank you for taking the time to review our paper. Let us here answer your questions.
>
> **[W1] Although the proposed estimator leverages diffusion bridge models, which differ from diffusion models, the proposed framework appears as an extension of MINDE**
>
> **Answer**: Dear reviewer, our methodology **is not an extension of MINDE**. It is based on a **different class of stochastic processes -  reciprocal processes**, which allows us to formulate the MI estimation task as domain translation rather than noise-data generative modeling, as in the case of MINDE. In addition, our method is theoretically unbiased, while MINDE method provides only biased MI estimation.
>
> Further, we refer the reviewer to the **Appendix C** “Conceptual comparison with MINDE” section.
>
> In addition, to prove the practical benefit of our InfoBridge w.r.t. MINDE, we refer the reviewer to our **[W3]** answer to reviewer **zM99**.
>
> **[W2] The paper is not well-written. The clarity could definitively be improved. For instance, SDE is not defined (stochastic differential equation). The equation in line 145 should be written in two lines, as it exceeds the borders of the paper. There are many typos: line 96 (conditioned its on values), line 264 (as a domain transfer task), multiple wrong references to equation (2) while I assume the authors wanted to refer to other equations. The same wrong reference to (2) happens also in line 300. Table 2 reports Mean Average Error, but I assume the authors used the Mean Absolute Error.**
>
> **Answer**: Thank you for noting some inaccuracies. We have added the SDE term to the notations section and reinforced it with a relevant reference. The equation on line 145 of the original submission has been rewritten in two lines, and references to it have been fixed.  The reference in line 300 is not a reference to an equation but rather a reference to the whole Background Section 2, including the Bridge Matching paragraphs. To be more precise in our references, we have changed it to the reference to equation 7 in the revised version. Indeed, in Table 2 we meant to report the  Mean Absolute Error rather than Mean Average Error, and we fixed all the MAE transcription throughout the paper. We’ve tracked the space constraints violation across the paper and fixed it.
>
> **[W3] In the experiments the authors compare the proposed method with f-DIME. However, there is no reference of f-DIME in the related work section, so it should be inserted in the Discriminative estimators paragraph.**
>
> **Answer:** Thank you for pointing this out. We have added the f-DIME reference to the related work section in the revised manuscript (Discriminative estimators paragraph, highlighted in blue).
>
> **[Q1] What is $\delta()$ in equation (3)? It is not defined.**
>
> **Answer:** The $\delta$ is the standard notation for the Dirac delta function. We have added a clarification to the Notations paragraph in the revised version.
>
> **[Q2] In line 148 the authors refer to Problem (2). Does it refer to the second equation? That would be the MI definition. In the line below the authors write “minimizes (2)”, but they are not referring to the minimization of MI, I guess they are referring to the minimization of the long equation in line 145.**
>
> **Answer:** We thank the reviewer for noticing the mismatch in referencing. Indeed, the reference is supposed to be for the long equation in line 145, which lacks the number. We have fixed all the references to this equation in the current revision.

---

> > ### Author Response · Authors · 2025-11-22
> > **Continuation of the rebuttal**
> >
> > **[Q3] In the pseudocode of algorithm 1 and in the code provided in the supplementary material, the authors perform a random permutation to obtain the samples from the product of marginal densities: y_samples_permuted = y_samples[torch.randperm(y_samples.shape[0])]. However, using such a function leads a certain number of elements of the shuffled vector to be positioned in the same initial position (before the permutation). This strategy was highlighted in f-DIME’s paper as problematic and in that specific case the authors showed that using a specific discriminative architecture that mistake would lead to an upper bound in the MI estimate. For InfoBridge I do not see any upper bound. However, performing such a random permutation is logically incorrect and the authors should perform the experiments after correcting such a mistake. How does the performance of the proposed algorithm change after this correction? If it does not change, why?**
> >
> > **Answer:** Thank you for your insightful suggestion. We acknowledge that random permutation can be problematic for discriminative MI estimators, as demonstrated in the f-DIME paper. For our generative method, however, the theoretical impact is less clear and remains a question for future analysis. Therefore, we do not consider the use of permutation to be a mistake, but rather a design choice to ensure a fair comparison with other methods that do not use derangements. Motivated by your feedback, we conducted additional experiments using derangements for sampling from $\Pi_{X_0} \otimes \Pi_{X_1}$.
> >
> > We show the experimental results on the high MI Smoothed Uniform distributions setup, described in  Section 5. In addition, we have have implemented the completely new $\pi(x_1)$ samples drawing strategy for sampling from $\Pi_{X_0} \otimes \Pi_{X_1}$. MAE means Mean Absolute Error and the ground truth MI is described in the top row.
> >
> > Three rows of results correspond to 1) Independent Permutation strategy we used in our experiments 2) Derangement corresponding to review suggestion [fDIME] 3) New data draw where we just draw new $x_1$ samples from the dataset.
> >
> > | Samples draw strategy | MAE | MI=10 | MI=20 | MI=40 | MI=80 |
> > | --- | --- | --- | --- | --- | --- |
> > | Independent Permutation | 12.24 | 7.01 | 14.32 | 28.89 | 50.82 |
> > | Derangement | 10.51 | 7.24 | 14.74 | 31.21 | 54.78 |
> > | New data draw | 10.62 | 7.31 | 14.76 | 30.06 | 55.38 |
> >
> > The results showed a **slight performance improvement** across all the setups, which is consistent with f-DIME's findings.
> >
> > In that light, we thank the reviewer for this valuable suggestion, and we have added a discussion about the different strategies of sampling the training data to the main text, Section 4.2, and **Appendix C.4**.
> >
> > **[Q4] Where are the experiments justifying the sentence in lines 314-315 about the additional binary input?**
> >
> > **Answer:** To empirically illustrate why one shared neural network backbone approach is indeed better than two neural network approaches, we add a **direct comparison in Appendix C of the revised version** of the paper and additionally provide it here.
> >
> > As Two Neural Networks approach, we employ completely different neural networks for the approximation of $v_{\text{joint}}$ and $v_{\text{ind}}$, while under Single Neural Network approach, we mean the method we used in our paper. We test both approaches on the Image Data benchmark Rectangle 16x16 case. The Two Neural Networks approach was trained under exactly the same hyperparameters as the Single Neural Network approach that we used everywhere in our paper. The only distinction between the methods is that joint and independent drifts are modeled by different neural networks, and during the training stage, both of these neural networks are trained.
> >
> > | Model | GT MI=0.1 | GT MI=1 | GT MI=2.5 | GT MI=5 | GT MI=7.5 | GT MI=10 |
> > | --- | --- | --- | --- | --- | --- | --- |
> > | Single Neural Network, MI estimation | 0.0 | 1.16 | 2.79 | 5.47 | 7.67 | 9.82 |
> > | Two Neural Networks, MI estimation | 0.83 | 1.86 | 3.49 | 5.93 | 8.12 | 10.23 |
> >
> > **Table**: In the top row, ground truth MI values are described, while the Single Neural Network and Two Neural Networks rows present the MI estimations by the corresponding method.
> >
> > One can see that the Two Neural Networks approach consistently overestimates the MI and generally provides more error. That is why we stick to the Single Neural Network approach.

---

> > > ### Author Response · Authors · 2025-11-22
> > > **Continuation of the rebuttal**
> > >
> > > **[Q5] Can the authors show a comparison of the computational complexity between the different algorithms used?**
> > >
> > > **Answer:** To summarize, the computational complexity of our algorithm as a Diffusion Bridge model is greater than that of discriminative methods and is similar to that of MINDE.  The runtime for each trainable method is already presented in **Appendix D** of the revised paper (and Appendix E of the original submission). We have added the related remarks at the end of **Section 4.2** of the revised version.
> > >
> > > **[Q6] Why the reference Lee and Rhee has no year?**
> > >
> > > **Answer:** We appreciate the reviewer noticing this oversight. We have changed it in the revised version.
> > >
> > > **[Q7] The discriminative models used during comparisons have the same size (number of parameters) of the generative estimators?**
> > >
> > > **Answer:** No, they do not. We found that discriminative estimators are more prone to overfitting than generative ones. Using equally complex networks for MINE-like estimators, in fact, yielded significantly worse results. Therefore,  to ensure a fair and meaningful comparison we either conducted a hyperparameter search to select an optimal, smaller architecture or used an already established one.
> > >
> > > The exact sizes of all the neural networks used are either explicitly described in the “Experimental supplementary” **Appendix D** of the revised version (Appendix E of the original submission) or referenced to another resource containing this information. To your convenience, we have updated **Appendix D** (Appendix E in the original submission) in the revised version and explicitly added all the sizes of all the neural networks we trained.
> > >
> > > **[Q8]  Isn’t 100k train samples a very large number? How do the algorithms perform with a lower number of training samples?**
> > >
> > > **Answer:** To answer your question, we have performed a study on the number of training samples. We present it in **Appendix C** of the revised version. In particular, we vary the **train sample size** in the Image Data experiment Section 5.2 with Rectangle images of 16x16 size. We test MINE, MINDE and InfoBridge on 5k, 10k, and 100k train sample sizes with a 10k test sample size. To prevent overfitting, we lower the number of parameters in the neural network for MINDE and InfoBridge. The neural network sizes are provided in the Table. All the other hyperparameters are exactly as previously. Methods with 5k and 10k train sample sizes are run with 1 seed, and 100k train sample size results were taken from the main paper. MAE means Mean Absolute Error. The best MAE for each training sample size is **bolded**.
> > >
> > > | Method | N samples | N params | MAE | 0.1 | 1 | 2.5 | 5 | 7.5 | 10 |
> > > | --- | --- | --- | --- | --- | --- | --- | --- | --- | --- |
> > > | InfoBridge (ours) | 5k | 106k | **0.199** | 0.008 | 0.79 | 2.33 | 4.83 | 7.2 | 9.75 |
> > > | MINDE | 5k | 106k | 2.397 | 0.47 | 0.36 | 0.44 | 0.38 | 4.9 | 5.91 |
> > > | MINE | 5k | 50k | 1.618 | 0.0 | 0.6 | 2.0 | 3.74 | 4.77 | 5.28 |
> > > | InfoBridge (ours) | 10k | 428k | **0.142** | 0.006 | 1.02 | 2.65 | 5.24 | 7.65 | 9.8 |
> > > | MINDE | 10k | 428k | 1.663 | 0.46 | 0.44 | 0.48 | 5.93 | 6.93 | 4.46 |
> > > | MINE | 10k | 50k | 1.513 | 0.0 | 0.78 | 2.1 | 3.9 | 4.85 | 5.39 |
> > > | InfoBridge (ours) | 100k | 27M | **0.228** | 0.0 | 1.16 | 2.79 | 5.47 | 7.67 | 9.82 |
> > > | MINDE | 100k | 27M | 0.815 | 0.41 | 0.91 | 2.45 | 4.22 | 5.63 | 8.21 |
> > > | MINE | 100k | 50k | 0.978 | 0.47 | 1.15 | 2.43 | 4.41 | 5.86 | 6.95 |
> > >
> > > One can see that our method retains its performance on a lower sample size, while MINDE and MINE start to degrade.
> > >
> > > **[Q9] In Figure 2 the authors specify which version of MINDE they are using, but do not specify which f-DIME version. Can the authors specify it?**
> > >
> > > **Answer:** The f-DIME version that we have shown in the main text is specified in **Appendix D** of the revised version (Appendix E of the original submission), i.e., the version with GAN divergence and Joint Architecture. We have added the explicit notion of f-DIME specification in the main text, **Section 5.4**.
> > >
> > > For completeness, all the other f-DIME formulations are tested in the **Appendix D** Tables 10-13 and we chose for comparison the best performing one.
> > >
> > > **Concluding remarks.** We would be grateful if you could let us know if the explanations we gave have been satisfactory in addressing your concerns and questions about our work. If so, we kindly ask that you consider increasing your rating. We are also open to discussing any other questions you may have.

---

> > > > ### Comment · Reviewer_uyVy · 2025-11-24
> > > >
> > > > Thank you for the detailed responses. I am willing to raise my score. However, before that, I have two final questions:
> > > > - Can you design a table with the runtime of the different algorithms tested in the paper?
> > > > - For your method, do you need to use a GPU? I read that you reported the GPU hours. Therefore, this should be inserted in the runtime comparison table, explicitly stating the hardware requirements to run the different methods.

---

> > > > > ### Author Response · Authors · 2025-11-25
> > > > > **Continuation of the rebuttal**
> > > > >
> > > > > In response to your questions, we have included runtime (approximate) and hardware comparisons for the Image Data Benchmark, Protein Data, and High MI experiments. These results are provided in **Appendix D** of the revised version, specifically in **Table 13**, **Table 15**, and **Table 20**.
> > > > >
> > > > > As noted, our method indeed utilizes a GPU, which is common among neural network based MI estimators. To ensure clarity and transparency, we have explicitly listed the hardware specifications for all compared methods in the corresponding tables.
> > > > >
> > > > > Here, we present one such table related to the Image data benchmark 16x16 setup. GPU states for NVIDIA A100, and CPU states for Intel Xeon Gold.
> > > > >
> > > > > | Method | Runtime | Hardware |
> > > > > | --- | --- | --- |
> > > > > | InfoBridge | 6 hours | GPU |
> > > > > | MINDE | 2 hours | GPU |
> > > > > | MIENF | 4 hours | GPU |
> > > > > | InfoNCE | 1 hour | GPU |
> > > > > | MINE | 1 hour | GPU |
> > > > > | NWJ | 1 hour | GPU |
> > > > > | KSG | 1 hour | CPU |
> > > > >
> > > > > We omit the low-dimensional benchmark runtime and hardware comparison table, because most of the results were taken from previous work, which do not report runtime and/or hardware for corresponding methods.  However, we report the runtime and required hardware for InfoBridge (ours) and NVF, JVF methods in **Appendix D**.

---

### Official Review · Reviewer_at8D · 2025-10-27

**Soundness:** 3
**Presentation:** 3
**Contribution:** 3
**Rating:** 6
**Confidence:** 4

**Summary:**

The authors propose InfoBRIDGE, a mutual information (MI) estimator that reframes MI estimation as a domain transfer problem between a joint distribution and the product of marginals using reciprocal processes built from mixtures of Brownian bridges. The required drifts are learned via conditional bridge matching with a neural net; training uses samples from the joint and a permuted batch to emulate independence, and evaluation computes the integral by Monte Carlo over Brownian-bridge samples. The authors argue this yields an unbiased MI estimator (given perfect drift learning) and report results on low-dimensional synthetic, image-based, high-MI benchmarks, and protein-embedding data.

**Strengths:**

* Conceptual novelty: Clear, principled decomposition of MI into a time-integral of drift differences between two bridge processes; different viewpoint than score-based MINDE.
* Methodological soundness: Builds on established conditional bridge matching; training and estimation procedures are straightforward (Alg. 1–2).
* Practicality: Uses only joint samples and Brownian-bridge simulation (no explicit density estimation), which is attractive for complex data like images/embeddings.

**Weaknesses:**

* Unclear dependence on ϵ: The estimator formula scales with 1/(2ϵ). Please analyze or empirically demonstrate ϵ-invariance of the resulting MI and give guidance for choosing ϵ.
* Bias/variance in practice: “Unbiased” is only in the idealized limit. Did you check the empirical calibration with confidence intervals vs. ground-truth MI? Did you check sensitivity to network size, training steps, and t-discretization?
* Independence construction: The independent process uses permuted pairs (x_0,\hat{x}_1). What is possible bias when datasets contain duplicated or highly correlated samples? Will a two-sample protocol (fresh marginal draws) materially changes results?
* Scope of experiments: The authors mention low-dimensional, image, high-MI, and protein settings; more detail is needed on metrics, baselines, and runtime/sample efficiency, especially versus strong discriminative bounds (e.g., InfoNCE variants).

**Questions:**

* Is MI provably independent of the choice of ϵ? in Theorem 4.1? If yes, please add a formal statement and a numeric ablation sweeping ϵ.
* What is the estimator variance as a function of t-grid resolution and batch size? Any control variates that reduce variance when t → 1 (where the (1-t)^{-1} factor amplifies noise)?
* Does training two drifts (joint vs. independent) materially increase sample complexity vs. one-model approaches? Could a shared backbone with binary conditioning stabilize learning?
* How would you share a NN between variables of different dimensionality?
* How does InfoBRIDGE behave on long-tailed or heavy-tailed marginals, beyond the cited remarks that classical estimators struggle there? Please add such stress tests.
* Table 1: What are the reported metrics? Please add to caption description of the metric presented in the table.
* Figure 1: why MI estimation undershoots for Gaussians (a,b) but not for rectangles (c,d)?
* Discussion of basic implementation, training, and hyper-parameters details should be included in the main body. Include discussion of the computational cost.

Comment:

* Limitations should be discussed in the main body, not only in the appendix.

---

> ### Author Response · Authors · 2025-11-22
> **Rebuttal**
>
> Dear reviewer **at8D**, we thank you for taking the time to review our paper. Let us here answer your questions.
>
> **[W1]**  **Unclear dependence on ϵ: The estimator formula scales with 1/(2ϵ). Please analyze or empirically demonstrate ϵ-invariance of the resulting MI and give guidance for choosing ϵ.**
>
> **[Q1]** **Is MI provably independent of the choice of ϵ? in Theorem 4.1? If yes, please add a formal statement and a numeric ablation sweeping ϵ.**
>
> **Answer**: Yes, it is provably independent of $\epsilon$. In Theorem 4.1 (Mutual Information decomposition), the $\epsilon$ parameter is included implicitly through processes $Q$. Theorem 4.1 statement is true for every $\epsilon > 0$, and the proof in **Appendix A** of the revised version (Appendix B of the original submission) doesn’t rely on any assumptions on $\epsilon$ except for strict positivity.
>
> To foster readability, we have added the note that $Q$ processes bear the $\epsilon$ hyperparameter inside of them, and we do not show that $\epsilon$ explicitly in notation.
>
> In addition, we have the ablation study in **Appendix C** of the revised version (Appendix D in the original submission), where we do the sweep along the volatility coefficient $\epsilon$ for the Image Data and Protein Data experiments and show the performance of our method w.r.t. different volatility coefficients, see **Tables 4 and 5**.  One can see that the results are comparable for different epsilon coefficients.
>
> **[W2]**  **Bias/variance in practice: “Unbiased” is only in the idealized limit. Did you check the empirical calibration with confidence intervals vs. ground-truth MI? Did you check sensitivity to network size, training steps, and t-discretization?**
>
> **Answer:** Yes, indeed, our method is “unbiased” only in the idealized limit; we note that in the footnote at the bottom of page 5.
>
> Empirically, we do several things to show that our method delivers robust MI estimation under changing settings.
>
> **First**, in the initial submission, we already delivered the confidence intervals for all the parametric MI estimation methods in the Image Data experiment. In particular, in **Figure 1**, we plot 99% confidence intervals w.r.t. MI estimation along the different seed runs, and in **Table 2**, one can see the std w.r.t. different seed runs averaged over all the setups. One can see from **Figure 1** that our confidence intervals are tight and from **Table 2** that our different seed runs’ std is among the best. While, for example, MINDE std is 6 times higher, and confidence intervals are a lot wider.
>
> **Second**, we check the sensitivity of our method w.r.t. neural network size and $\epsilon$ hyperparameter in **Appendix C.2** of the revised version (Appendix D.2 in the original submission).
>
> Since we have continuous time $t$ we do not have a t-discretization strategy and just sample $t$ uniformly.
>
> **Third**, upon request in **Appendix C.2** of the revised paper version, we analyze the robustness of our method w.r.t. number of training steps and present the MI estimation dynamics during training. As one can see in **Table 7** (duplicated here below) our method is robust w.r.t. number of training iterations.
>
> N epochs stands for the number of training epochs and GT MI stands for ground truth mutual information value. The closer to GT MI, the better. The experiment is held on High MI test with Smoothed Uniform distribution (see Section 5.4). All the other hyperparameters are the same as in the Section 5.4 experiment.
>
> As one can see, **our method is robust** w.r.t. number of **training iterations/epochs**.
>
> | N epochs\ GT MI | 10 | 20 | 40 | 80 |
> | --- | --- | --- | --- | --- |
> | 10 | 8.14 | 16.09 | 30.87 | 47.7 |
> | 20 | 7.79 | 15.51 | 30.26 | 48.61 |
> | 30 | 7.7 | 15.5 | 30.47 | 50.16 |
> | 60 | 7.51 | 15.02 | 29.67 | 50.2 |
> | 120 | 7.29 | 14.63 | 29.55 | 50.01 |
> | 200 | 7.01 | 14.32 | 28.89 | 50.82 |
>
> In addition, in the **Appendix C.3** of the revised version, we also include the study on our InfoBridge method robustness w.r.t. **number of training samples**, which is very important on practice. One can see that in **Table 8** that our method works well even on a **20x times lower training sample size**.

---

> > ### Author Response · Authors · 2025-11-22
> > **Continuation of the rebuttal**
> >
> > **[W3] Independence construction: The independent process uses permuted pairs (x_0,\hat{x}_1). What is possible bias when datasets contain duplicated or highly correlated samples? Will a two-sample protocol (fresh marginal draws) materially changes results?**
> >
> > **Answer**: Thank you for your insightful suggestion.  Following your feedback, we have analyzed various strategies for sampling the data points for learning the independent process and conducted additional experiments shown in **Appendix C** of the revised paper. For the reviewer's convenience, we have added experimental results and a discussion to our answer below.
> >
> > We consider several strategies for drawing training samples. The **Independent Permutation** strategy we used in our experiments. **Derangement** strategy proposed in [fDIME] to solve exactly the problem of correlation between samples after the independent permutation. This approach generates the permutation where no item stays in the same place. Suggested by the reviewer fresh sample draw.
> >
> > The experimental results are shown on the high MI Smoothed Uniform distributions setup, described in Section 5. MAE means Mean Absolute Error, and the ground truth MI is described in the top row.
> >
> > | Samples draw strategy | MAE | MI=10 | MI=20 | MI=40 | MI=80 |
> > | --- | --- | --- | --- | --- | --- |
> > | Independent Permutation | 12.24 | 7.01 | 14.32 | 28.89 | 50.82 |
> > | Derangement | 10.51 | 7.24 | 14.74 | 31.21 | 54.78 |
> > | Fresh sample draw | 10.62 | 7.31 | 14.76 | 30.06 | 55.38 |
> >
> > The results showed a **slight performance improvement** for both derangement and fresh marginal draws strategies across all the setups.
> >
> > In that light, we thank the reviewer for this valuable suggestion, and we have added a discussion about the different strategies of sampling the training data to the main text, **Section 4.2**, and **Appendix C.3**.
> >
> > **[W4]** **Scope of experiments: The authors mention low-dimensional, image, high-MI, and protein settings; more detail is needed on metrics, baselines, and runtime/sample efficiency, especially versus strong discriminative bounds (e.g., InfoNCE variants).**
> >
> > **[Q8]** **Discussion of basic implementation, training, and hyperparameter details should be included in the main body. Include a discussion of the computational cost.**
> >
> > **Answer:** As the main metric, we use just MI estimations of the method and ground truth MI, which is an established practice in evaluating MI estimation methods **[MINE, MINDE, fDIME]**. In addition, we utilize the MAE and std discussion on which are presented in **Appendix D.1** of the revised paper version.
> >
> > All the other details, including training, runtime, and all the baseline’s hyperparameters, are described in **Appendix D** of the revised version (Appendix E of the original submission). The computational cost of each method is described in **Appendix D** of the revised version (Appendix E of the original submission). We have added a statement that our method has computational complexity similar to MINDE in the main text, **Section 4.2**.
> >
> > Due to the paper space constraints, it is not possible to include all the experimental details in the main text. The **basic implementation** is described in **Algorithms 1 and 2**. A lot of details are described in **Section 5** itself, and we have moved some of the hyperparameters **from the Appendix to the main text** in the revised version.
> >
> > Additionally, we have shown the sample efficiency of our method in **Appendix C.3** Ablation study.
> >
> > **[Q2]** **What is the estimator variance as a function of t-grid resolution and batch size? Any control variates that reduce variance when t → 1 (where the (1-t)^{-1} factor amplifies noise)?**
> >
> > **Answer**: We have a continuous time variable $t$ and hence we do not have any t-grid. To control the variance during learning, we follow regular Bridge Matching methodology and sample the $t \sim U[0, 1 - delta]$, where delta is very small, i.e., 1e-03.

---

> > > ### Author Response · Authors · 2025-11-22
> > > **Continuation of the rebuttal**
> > >
> > > **[Q3]** **Does training two drifts (joint vs. independent) materially increase sample complexity vs. one-model approaches? Could a shared backbone with binary conditioning stabilize learning?**
> > >
> > > **Answer:** You are absolutely right in raising the question about drift parameterizations and one-two model approaches. In fact, we **already** use only **one neural network**, and describe our approach in the paragraph “Vector field parametrization” in **Section 4.2**. In particular, we use only **one neural network** with binary conditional input to handle both joint and independent drifts. In that light, we do not face potential issues related to two neural networks.
> > >
> > > To empirically illustrate why one shared neural network backbone approach is indeed better than two neural network approaches, we add a **direct comparison in Appendix C.5 of the revised version** of the paper and additionally provide it here. We test both approaches on **Image Data benchmark Rectangle 16x16** case. The Two Neural Networks approach was trained under exactly the same hyperparameters as the Single Neural Network approach that we used everywhere in our paper. The only distinction between the methods is that joint and independent drift are modeled by different neural networks, and during the training stage, both of these neural networks are trained.
> > >
> > > | Model | GT MI=0.1 | GT MI=1 | GT MI=2.5 | GT MI=5 | GT MI=7.5 | GT MI=10 |
> > > | --- | --- | --- | --- | --- | --- | --- |
> > > | Single Neural Network, MI estimation | 0.0 | 1.16 | 2.79 | 5.47 | 7.67 | 9.82 |
> > > | Two Neural Networks, MI estimation | 0.83 | 1.86 | 3.49 | 5.93 | 8.12 | 10.23 |
> > >
> > > **Table**: In the top row, ground truth MI values are described, while the Single Neural Network and Two Neural Networks rows present the MI estimations by the corresponding method.
> > >
> > > One can see that the Two Neural Networks approach consistently overestimates the MI and generally provides more error. That is why we stick to the Single Neural Network approach.
> > >
> > > **[Q4]** **How would you share a NN between variables of different dimensionality?**
> > >
> > > **Answer:** In our discussion on the estimation of MI between variables in **Appendix B.5** of the revised version (Appendix C.5 of the original version) we note several methods to perform this. In practice, each of the proposed methods leads us to an equivalent formulation of MI estimation in terms of variables of the **same dimension**. In that case, we **do not need to share the neural networks between variables of different dimensionality**.
> > >
> > > **[Q5]** **How does InfoBRIDGE behave on long-tailed or heavy-tailed marginals, beyond the cited remarks that classical estimators struggle there? Please add such stress tests.**
> > >
> > > **Answer:** We do already have these tests in our paper in the low-dimensional benchmark experiment **Table 1**, in particular **student distributions** st 1 x 1, st 2 x 2, e.t.c., and we explore the heavy-tailed **half cube transform** applied for Gaussian random variable in the High Mutual Information experiment **Figure 2**.
> > >
> > > **[Q6]** **Table 1: What are the reported metrics? Please add to caption description of the metric presented in the table.**
> > >
> > > **Answer:** As indicated in the caption of Table 1: “Mean MI estimates over $ 10 $ seeds using $ 10 $k test samples against ground truth (GT)”. Therefore, in Table 1, the MI estimation is reported. The closer to the ground truth, the better. We have added additional remarks to the Table 1 caption in the revised version.
> > >
> > > **[Q7]** **Figure 1: why MI estimation undershoots for Gaussians (a,b) but not for rectangles (c,d)?**
> > >
> > > **Answer:** The Gaussian blobs dataset has an intrinsic dimensionality of 2, while the rectangles dataset has a dimensionality of 4. Consequently, for the same ground-truth MI, the MI per latent dimension is higher for the Gaussians. This leads to minor numerical instabilities, especially at high MI values where the relationship is almost deterministic and thus challenging for neural estimators to capture. This is consistent with the increased undershooting observed in other methods under the same conditions.
> > >
> > > **[Comment 1]**  Limitations should be discussed in the main body, not only in the appendix.
> > >
> > > **Answer:** Following your advice, we have added the limitations section to the main body in the revised version.
> > >
> > > **Concluding remarks.** We would be grateful if you could let us know if the explanations we gave have been satisfactory in addressing your concerns and questions about our work. If so, we kindly ask that you consider increasing your rating. We are also open to discussing any other questions you may have.

---

> > > > ### Author Response · Authors · 2025-11-22
> > > > **Continuation of the rebuttal**
> > > >
> > > > **Citations:**
> > > >
> > > > [MINE]  Mohamed Ishmael Belghazi, Aristide Baratin, Sai Rajeshwar, Sherjil Ozair, Yoshua Bengio, Aaron Courville, and Devon Hjelm. Mutual information neural estimation. In Jennifer Dy and Andreas Krause (eds.), Proceedings of the 35th International Conference on Machine Learning, volume 80 of Proceedings of Machine Learning Research, pp. 531–540. PMLR, 07 2018
> > > >
> > > > [MINDE] Giulio Franzese, Mustapha BOUNOUA, and Pietro Michiardi. MINDE: Mutual information neural diffusion estimation. In The Twelfth International Conference on Learning Representations, 2024.
> > > >
> > > > [fDIME]  Nunzio Alexandro Letizia, Nicola Novello, and Andrea M Tonello. Mutual information estimation via f-divergence and data derangements. Advances in Neural Information Processing Systems, 37: 105114–105150, 2024.

---

### Official Review · Reviewer_zM99 · 2025-11-05

**Soundness:** 4
**Presentation:** 2
**Contribution:** 3
**Rating:** 4
**Confidence:** 5

**Summary:**

This paper propose an unbiased estimator of mutual information constructed on bridge matching.
The evaluation is solely conducted by measuring the discrepancy from a ground truth mutual information.

**Strengths:**

+ The main contribution is much sought after: An unbiased, low variance estimate of mutual information has impact across fields.

+ The paper's literature review is apt and does a great job motivating the problem.

+ The paper is full of illuminating insights tying information theory, diffusion models and stochastic calculus.

**Weaknesses:**

+ The evaluation is solely conducted by measuring the discrepancy from a ground truth mutual information. This is sound an necessary but also limited by itself. The paper would be more convincing if the proposed estimate of the mutual information were
used as an objective, or its learned representations evaluated on downstream tasks.

+ The work could be made much more accessible. The background section runs two full pages, and the authors only introduce their method in page 5. We suggest the more direct route of introducing a simple example right after the introduction, and introduce the stochastic calculus concepts as needed.

+ The paper differentiate itself from MINDE[1] by 1) being unbiased while the latter is only asymptotically unbiased, 2) By presenting itself as a domain transfer task. Unfortunately, this shift in conceptual framing while thought provoking is only  indirectly evaluated through a reduction of the estimators in variance. This is a bit unfortunate as eliminating bias and reducing variance is enough to differentiate well enough from MINDE.

[1] Franzese, Giulio, Mustapha Bounoua, and Pietro Michiardi. "MINDE: Mutual information neural diffusion estimation." arXiv preprint arXiv:2310.09031 (2023).

**Questions:**

The typical condition for the application of Girsanov's theorem is that of Novikov. How do you suggest relaxing it to use InfoBridge with long tailed distributions as suggested in section 6?

---

> ### Author Response · Authors · 2025-11-22
> **Rebuttal**
>
> Dear reviewer **zM99**, we thank you for taking the time to review our paper. Let us here answer your questions.
>
> **[W1] The evaluation is solely conducted by measuring the discrepancy from a ground truth mutual information. This is sound an necessary but also limited by itself. The paper would be more convincing if the proposed estimate of the mutual information were used as an objective, or its learned representations evaluated on downstream tasks.**
>
> **Answer**: Thank you for this suggestion. While our original submission did not explore the InfoMax self-supervised representation learning (SSRL) direction, we agree that it is a relevant area for evaluation. However, as noted in prior work [Tschannen’20], higher mutual information estimates do not necessarily correlate with higher-quality embeddings; in fact, simpler estimators often yield superior results in such InfoMax settings [Figures 1-3 in Tschannen’20].
>
> Nevertheless, motivated by your suggestion, we adopted a Deep InfoMax-like self-supervised learning framework from [Butakov’25]. We selected this work because it directly links mutual information to embedding quality and distribution. Consequently, we retain the InfoNCE objective as a computationally efficient surrogate for MI maximization, while using *Info*Bridge to evaluate the final quality of the learned representations. In particular, we utilize the MNIST data experimental setup from [Butakov’25] and learn the embeddings for 500 epochs.
>
> The MI estimation of **InfoBridge** is **closer** to the original capacity of the channels **than** the **InfoNCE** MI estimation. That demonstrates that the quality and distribution of the embeddings are better than expected. One can see the particular results in the following table, where the top row demonstrates the dimensionality of embeddings and the other rows demonstrate MI estimations and the capacity of the channels in nats.
>
> | Embedding DIM | 2 | 4 | 8 | 16 | 32 |
> | --- | --- | --- | --- | --- | --- |
> | Capacity | 4.61 | 9.21 | 18.42 | 36.84 | 73.68 |
> | InfoNCE MI estimation | 3.96 | 6.04 | 6.21 | 6.21 | 6.22 |
> | InfoBridge MI estimation | 3.95 | 8.23 | 13.13 | 22.14 | 30.5 |
>
> **[W2] The work could be made much more accessible. The background section runs two full pages, and the authors only introduce their method in page 5. We suggest introducing a simple example immediately after the introduction, and then introducing the stochastic calculus concepts as needed.**
>
> **Answer:** We deeply appreciate the reviewers’ comments on the improvement of our paper presentation. The reciprocal processes and bridge matching details described in the Background are crucial for the understanding of the method and make the methodology in Section 4 a lot lighter. In that light, it would be difficult to remove the mentioned parts from the Background section.
>
> However, following your comment, we have made the background a bit lighter and have shifted one part related to stochastic calculus to the main methodological part. We hope that the change makes our paper more accessible and concise. You can take a look at changes in the revised version of the paper (Sections 2 and 4).

---

> > ### Author Response · Authors · 2025-11-22
> > **Continuation of the rebuttal**
> >
> > **[W3]  The paper differentiate itself from MINDE[1] by 1) being unbiased while the latter is only asymptotically unbiased, 2) By presenting itself as a domain transfer task. Unfortunately, this shift in conceptual framing while thought provoking is only indirectly evaluated through a reduction of the estimators in variance. This is a bit unfortunate as eliminating bias and reducing variance is enough to differentiate well enough from MINDE.**
> >
> > **Answer**: If we correctly understand you, while there are strong enough theoretical differences between our InfoBridge and competitor approach MINDE the practical difference in performance lies only in the reduction in MI estimation variance. To begin with, we notice that the conceptual discussion on differences between MINDE and InfoBridge is held out in **Appendix C** in the revised version (Appendix D of the original submission). To further address this, we would like to **extrapolate on the practical differences** between our InfoBridge and MINDE.
> >
> > We think that the reduction of algorithm variance is a significant contribution. In general, the unbiasedness and domain transfer formulation allow us to have a more stable and robust algorithm. The stability and reduced variance manifest in several ways:
> >
> > - InfoBridge has **10x times less variance** in MI estimation on Image Benchmark data than MINDE. For detailed analysis see **Appendix C Table 3** of the revised version (**Appendix D** of the original submission).
> > - Stable optimization in protein data experiment that leads to **200x and 33000x times less error** than MINDE. For detailed analysis see **Appendix D.4 Table 13** of the revised version (Appendix E of the original submission).
> > - **Stability** w.r.t. **lowering the train samples size** on Image Benchmark data. Detailed analysis is included in **Appendix C** of the revised version and is **duplicated lower**.
> >
> > In addition, our method **estimates MI more precisely** than MINDE in almost all the experiments.
> >
> > We believe that this is enough to show the viability of our InfoBridge approach in practice.
> >
> > **Lower train samples size ablation**
> >
> > Here, we claim that our method is robust w.r.t. **train sample size variability** while MINDE isn’t. Next, we present the Image Data experiment **Section 5.2** with Rectangle images of 16x16 size. We test the method under lower train dataset size, i.e., 5k and 10k samples, instead of 100k used in the main text. To prevent overfitting we lower the number of parameters in the neural network for InfoBridge and MINDE. Other experiment details are the same as in the main paper. For completeness, we also add the MINE method to the comparison.
> >
> > | Method | N samples | N params | MAE | 0.1 | 1 | 2.5 | 5 | 7.5 | 10 |
> > | --- | --- | --- | --- | --- | --- | --- | --- | --- | --- |
> > | InfoBridge (ours) | 5k | 106k | **0.199** | 0.008 | 0.79 | 2.33 | 4.83 | 7.2 | 9.75 |
> > | MINDE | 5k | 106k | 2.397 | 0.47 | 0.36 | 0.44 | 0.38 | 4.9 | 5.91 |
> > | MINE | 5k | 50k | 1.618 | 0.0 | 0.6 | 2.0 | 3.74 | 4.77 | 5.28 |
> > | InfoBridge (ours) | 10k | 428k | **0.142** | 0.006 | 1.02 | 2.65 | 5.24 | 7.65 | 9.8 |
> > | MINDE | 10k | 428k | 1.663 | 0.46 | 0.44 | 0.48 | 5.93 | 6.93 | 4.46 |
> > | MINE | 10k | 50k | 1.513 | 0.0 | 0.78 | 2.1 | 3.9 | 4.85 | 5.39 |
> > | InfoBridge (ours) | 100k | 27M | **0.228** | 0.0 | 1.16 | 2.79 | 5.47 | 7.67 | 9.82 |
> > | MINDE | 100k | 27M | 0.815 | 0.41 | 0.91 | 2.45 | 4.22 | 5.63 | 8.21 |
> > | MINE | 100k | 50k | 0.978 | 0.47 | 1.15 | 2.43 | 4.41 | 5.86 | 6.95 |
> >
> > One can see that our method's performance stays almost the same, while MINDE's performance degrades. We have added this experiment to **Appendix C.3** of the revised version.
> >
> > **[Q1] The typical condition for the application of Girsanov's theorem is that of Novikov. How do you suggest relaxing it to use InfoBridge with long tailed distributions as suggested in section 6?**
> >
> > **Answer**: We thank the reviewer for this remark. Indeed, the Novikov condition seems not to be satisfied in most of the heavy-tailed diffusion cases, including the alpha-stable processes described in the referenced paper [Yoon’23]. We have removed this remark from Section 6 of the revised version.
> >
> > **Concluding remarks.** We would be grateful if you could let us know if the explanations we gave have been satisfactory in addressing your concerns and questions about our work. If so, we kindly ask that you consider increasing your rating. We are also open to discussing any other questions you may have.
> >
> > **Citations**:
> >
> > [Yoon’23] Eun Bi Yoon, Keehun Park, Sungwoong Kim, and Sungbin Lim. Score-based generative models with l´evy processes. Advances in Neural Information Processing Systems
> >
> > [Tschannen’20] M. Tschannen et al. “On Mutual Information Maximization for Representation Learning.” Proc. of ICLR 2020.
> >
> > [Butakov’25] I. Butakov et al. “Efficient Distribution Matching of Representations via Noise-Injected Deep InfoMax.” Proc. of ICLR 2025

---

### Author Response · Authors · 2025-11-22
**General Answer**

We sincerely thank all the reviewers for their valuable feedback and for recognizing the strengths of our work:

- Conceptual novelty and theoretical soundness of the approach (**ZM99, at8D, uyVy, UxJn**)
- Rich experimental evaluation including a set of benchmarks and ablation studies (**UxJn**)
- Practical effectiveness of the proposed approach  (**at8D, uyVy**)

In response to reviewers' concerns and suggestions, we have made the following **changes, highlighted in blue in the revised manuscript**:

- Fixed the misprints and added some additional notational clarifications (**uyVy**, **UxJn**)
- Shifted the Limitations section from the Appendix to the main text. Due to space constraints, it was not previously possible. (**at8D**)
- Provided full assumptions list in Appendix A.1 (**UxJn**)
- The following additional ablation studies and discussions have been added to Appendix C of the revised version: (**zM99, at8D, uyVy, UxJn**)
    - Robustness w.r.t. train sample size.
    - Robustness w.r.t. number of training iterations.
    - Discussion on strategies for sampling the independent data pairs for learning the $v_{\text{ind}}$ drift function.
    - Discussion on possible neural network parametrizations for drift functions.
- Added runtime comparison tables for all the reproduced methods in Appendix D (**uyVy**)
- And some other minor changes (**zM99, at8D, uyVy, UxJn**)

We believe that we have addressed the reviewers’ concerns and strengthened our manuscript through revision.

---

### Author Response · Authors · 2025-12-03
**Rebuttal update**

We once again sincerely thank all the reviewers for their valuable feedback. We believe that we have addressed all reviewer comments during the rebuttal phase and have submitted a revised version of the paper that includes cosmetic improvements, additional clarifications, and new ablation studies.

During the rebuttal stage, **two reviewers responded (at8D, uyVy)** to our rebuttals, and **both** were **positive** and expressed their willingness to **raise the score**.

---

Here we provide the **main condensed set of reviewer questions** and our answers:

| Question | Reviewers | Answer |
| --- | --- | --- |
| Utility of our method in learning latent representations? | **zM99** | We did an experiment applying InfoBridge for evaluation of final quality of learned representations in Deep InfoMax method. See answer to reviewer  **zM99**. |
| Comparison of our approach with MINDE? | **zM99, uyVy** | Theoretically, our method and MINDE are built on **different foundations**, and details are thoroughly described in **Appendix C.1.** In practice, our method delivers **no bias**, **less variance**, **stability w.r.t. number of train samples** and **less MI estimation error** on most of the tested problems. For details see answers to reviewers **zM99, uyVy.** |
| Theoretical and practical dependence on $\epsilon$? Guidance for choosing it? | **at8D, UxJn** | Our method is theoretically valid for every $\epsilon>0$ and the performance with different $\epsilon$ is checked empirically in the **Appendix C.2**. In addition we provide an intuition on how to choose $\epsilon$ in the answer to reviewer **UxJn.** |
| Sensitivity of our method w.r.t. different hyperparameters? | **at8D, uyVy** | We provide all the the ablation studies in **Appendix C** and did some additional ones by request (train samples size, number of training iterations, drift function parametrizations). |
| Construction of independent samples during training phase | **at8D, uyVy** | We ablate three different methods for sampling the independent pairs during the training of independent drift functions and show the results in **Appendix C.4.** Proposed be reviewers strategies deliver **slight performance improvement**. |
| Additional experimental details | **at8D, uyVy** | We have added all the requested details to **Appendix D**. |
| Misprints, notational clarifications, more accessible work? | **uyVy, zM99** | We have fixed all the misprints, formatting issues and clarified some notation. To make work more accessible we've shifted part of the background to the methodological section. |
| Training samples size study | **uyVy, UxJn** | We provide train samples size study and add it to **Appendix C.3**. As a conclusion our method is **robust w.r.t. lowering the number of samples in train dataset**. |
| Assumptions clarifications | **UxJn** | We have added an explicit list of assumptions in the **Appendix A.1.** |

---

### Meta-Review · Area_Chair_Z6vu · 2026-01-17

**Summary:**

In the words of `at8D`, this manuscript appears "conceptually novel" and "methodologically sound". It improves upon mutual information estimation, which, generally, is a difficult task. `UxJn` notes that it is thoroughly (though not at all times clearly) explained in its differences from the previous art, and theoretically well grounded.

**Reviewer Concerns:**

`zM99` notes that the work could be made much more accessible, a concern echoed by `uyVy`. While the authors have made some edits, even for someone familiar with the field the manuscript is dense.

`zM99` also notes that a primary use of these estimators is as a optimization target. This concern remains unaddressed.

**Reviewer Scores:**

Probably no changes.

---

### Decision · Program_Chairs · 2026-01-26

Accept (Poster)